# Primary production sensitivity to phytoplankton light attenuation parameter increases with transient forcing

Karin F. Kvale[1] and Katrin J. Meissner[2,3]

[1]GEOMAR Helmholtz-Zentrum für Ozeanforschung Kiel, Düsternbrooker Weg 20, D-24105 Kiel, Germany
[2]Climate Change Research Centre, Level 4 Mathews Building, UNSW, Sydney, NSW, Australia
[3]ARC Centre of Excellence for Climate System Science

*Correspondence to:* K. Kvale (kkvale@geomar.de)

**Abstract.** Treatment of the underwater light field in ocean biogeochemical models has been attracting increasing interest, with some models moving towards more complex parameterisations. We conduct a simple sensitivity study of a typical, highly simplified parameterisation. In our study, we vary the phytoplankton light attenuation parameter over a range constrained by data during both pre-industrial equilibrated and future climate scenario RCP8.5. In equilibrium, lower light attenuation parameters (weaker self-shading) shift net primary production (NPP) towards the high latitudes while higher values of light attenuation (stronger shelf-shading) shift NPP towards the low latitudes. Climate forcing magnifies this relationship through changes in the distribution of nutrients both within and between ocean regions. Where and how NPP responds to climate forcing can determine the magnitude and sign of global NPP trends in this high $CO_2$ future scenario. Ocean oxygen is particularly sensitive to parameter choice. Under higher $CO_2$ concentrations, two simulations establish a strong biogeochemical feedback between the Southern Ocean and low latitude Pacific that highlights the potential for regional teleconnection. Our simulations serve as a reminder that shifts in fundamental properties (e.g., light attenuation by phytoplankton) over deep time have the potential to alter global biogeochemistry.

## 1 Introduction

Treatment of marine light availability for photosynthesis in biogeochemical compartments of ocean general circulation models (OGCMs) has largely avoided careful scrutiny until recently (e.g., Dutkiewicz et al. 2015; Kim et al. 2015; Gregg and Rousseaux 2016). These models typically use simplified, empirically-based parameterisations of phytoplankton growth rates related to photosynthetically available radiation (PAR) based on the state of the science in the 1970s and 1980s. The OGCM in the University of Victoria Earth System Climate Model (UVic ESCM; Weaver et al. 2001; Eby et al. 2009) is one example. In it, the irradiance ($I$) at each depth level is calculated as:

$$I = I_{z=0}\text{PAR}\exp\left(-k_w\tilde{z} - k_c \int_0^{\tilde{z}} (P + Diaz)dz\right) \cdot [1 + a_i(\exp(-k_i(h_i + h_s)) - 1)] \tag{1}$$

(Schmittner et al., 2005; Kirk, 1983) where PAR stands for photosynthetically available radiation, $k_w$, $k_c$, and $k_i$ are light attenuation coefficients for water, phytoplankton (diazotrophs and general phytoplankton), and ice, $\tilde{z}$ is the effective vertical

coordinate, $a_i$ is the fractional sea ice cover, and $h_i$ and $h_s$ are calculated sea ice and snow cover thicknesses. Phytoplankton concentration ($P$ and $Diaz$) is expressed in a base unit of mmol N m$^{-3}$. Light attenuation coefficients $k_w$ and $k_i$ have the unit of m$^{-1}$, but light attenuation by phytoplankton is dependent on phytoplankton concentration (Kirk, 1975) and $k_c$ is expressed in units of (m mmol N m$^{-3}$)$^{-1}$. However, $k_c$ cannot be considered to represent the light attenuation of phytoplankton only, but

also represents the attenuation of constituents that are assumed to co-vary with phytoplankton (i.e., particulate and dissolved inorganic and organic materials). Light attenuation coefficients are classified as apparent optical properties (AOP) because they represent the combined effect of the inherent optical properties (IOP) of the medium (e.g., seawater or phytoplankton cells) and the optical properties of the radiation field (see Kirk, 1983; Falkowski and Woodhead, 1992, and associated references). Early tests of $k_c$ (e.g., Fasham et al., 1990) demonstrated low model biomass sensitivity to parameter value choice, and this has been

the prevailing wisdom of biogeochemical modellers for over 20 years. Estimates of $k_c$ vary widely: for example, 0.014 m$^2$(mg Chl $a$)$^{-1}$ (generally applicable, Lorenzen, 1972), 0.041 m$^2$(mg Chl $a$)$^{-1}$ (Southern Ocean, Bracher and Tilzer, 2001), or a range from 0.006 to 0.015 m$^2$(mg Chl $a$)$^{-1}$ assuming all phytoplankton represent mixes of specific species of dinoflagellates, calcifiers, or diatoms (Falkowski et al., 1985). Even the assumption that $k_c$ varies predictably with chlorophyll concentration can be considered highly simplistic because the co-varying constituents might cause this ratio to fluctuate (Siegel et al., 2005).

In practice, any value assigned to $k_c$ is going to be highly model-dependent (e.g., 0.058 m$^2$(mg Chl $a$)$^{-1}$ in Wang et al., 2008) because of the wide range of observational estimates and the necessary conversion from chlorophyll to model nutrient units, which requires some assumptions that depend on model structure.

While it was recognized early on that a rigorous description of the spectral composition of the underwater light field must separate effects of IOP from the radiation field, early measurements emphasized AOP because of technological limitations as

well as for a lack of data resolving the IOP of seawater constituents (Kirk, 1983). Research into the IOP of these constituents has since benefited from continuously improving analytical tools including satellite remote sensing, whose algorithms depend on their resolution (Sathyendranath and Platt, 2007). Phytoplankton IOP are species-specific (e.g., Stramski et al., 2001). Of the major seawater constituents, detrital particles contribute the most to light scattering and attenuation, and picoplankton are the primary phytoplankton light absorbers (Stramski et al., 2001; Fujii et al., 2007), though their contribution to absorption is

possibly exceeded by coloured dissolved organic matter (CDOM) (Siegel et al., 2005; Fujii et al., 2007).

Recent work has highlighted what we potentially miss in our OGCMs, and hence our earth system models (ESMs) as well, by neglecting explicit radiative transfer and IOP. Decomposing the calculation of underwater irradiance into IOP, resolving a variable number of radiative wavebands, and then testing three parameterisations of light limitation of phytoplankton growth in a one dimensional ecosystem model comparable to observational data show that a model resolving multiple wavebands

and a spectral sensitivity in photosynthesis outperformed a simple parameterisation similar to Equation 1 (Kettle and Merchant, 2008). Choice in parameterisation of spectral resolution can contribute percentage errors of up to 200% (Kettle and Merchant, 2008). Not including a radiative transfer model in an OGCM can reduce global nitrate by 33% and chlorophyll by 24%, and bring about large changes in phytoplankton biogeography, even when there is no change in total irradiance (Gregg and Rousseaux, 2016). Dutkiewicz et al. (2015) offer the most complicated model analysis, and demonstrate that explicitly

resolving radiative transfer and the IOP of phytoplankton types, detritus, and CDOM in a three-dimensional ocean model can

improve model performance against observational data. Their sensitivity analysis demonstrates complex feedbacks between light, phytoplankton attributes, and biogeochemistry (Dutkiewicz et al., 2015).

Explicit radiative transfer and fully resolved IOP add computational expense to already expensive ESMs. Since empirically-based AOP parameterisations are still widely used and economical, it is useful to test their limitations and sensitivities. Including light attenuation by phytoplankton in an OGCM amplifies the seasonal cycle of temperature, mixed layer depth and ice cover by about 10% over neglecting it altogether (Manizza et al., 2005). Gnanadesikan and Anderson (2009) find the inclusion of light attenuation by chlorophyll in an OGCM alters physical water mass characteristics including a decrease in subsurface temperatures by over 2°C in the low latitudes. Kim et al. (2015) explore the biogeochemical consequences of differentiating light attenuation by CDOM and detrital particles from that of chlorophyll in an ESM and find these components increased surface phytoplankton biomass by reducing levels of light at depth, which reduced deeper production and made more nutrients available at the surface. Two model simulations with-and-without CDOM and detrital particle light attenuation differ by 9% in the global average biomass and 7% in the global carbon export flux at 200 meters depth (Kim et al., 2015). These are modest changes with respect to other production and export parameters (e.g., Kwon et al. 2009 found a 5 Gt C y$^{-1}$, or 50%, increase in global carbon export by raising the export transfer efficiency exponent by 0.4), though regional sensitivities are stronger (Kim et al., 2015).

All of the studies mentioned above make their comparisons using models to which pre-industrial forcings are applied. Dutkiewicz et al. (2015) highlighted the potential for complex feedbacks arising due to model treatment of light and optical properties, therefore it stands to reason such feedbacks may compound under climate change. A recent bug fix in the implementation of Equation 1 in the UVic ESCM prompted a hasty equilibrated model re-evaluation, which then led to our more thorough assessment including climate change. Future implementation of a more complex radiative transfer and phytoplankton IOP model may be justified based on the conclusions of the authors above (e.g., Dutkiewicz et al., 2015), however the UVic ESCM (and other models of similar structure) is widely used in its current form and it is therefore worthwhile to assess and report on its current sensitivities. The aim of our study is to assess the sensitivity of modelled net primary production to phytoplankton light attenuation parameter value in an ESM using pre-industrial equilibrated, historical, and projected climate forcing. The drivers of net primary production are of scientific interest as they may respond to anthropogenic climate change (e.g., Kvale et al., 2015; Laufkötter et al., 2015). To our knowledge, such a simple assessment has not appeared in the peer-reviewed domain despite there being a wide range of phytoplankton light attenuation parameter values currently in use (described in more detail below) and a demonstrated sensitivity of primary production, export, and nutrients in OGCMs and ESMs to how the underwater light field is modelled (described above).

## 2 Methods

The University of Victoria Earth System Climate Model (UVic ESCM, Weaver et al. 2001, Eby et al. 2009) version 2.9 is a coarse-resolution ($1.8° \times 3.6° \times 19$ ocean depth layers) ocean-atmosphere-biosphere-cryosphere-geosphere model. The biogeochemical compartment (Schmittner et al., 2005, 2008; Keller et al., 2012) is a nutrients-phytoplankton-zooplankton-detritus

(NPZD) model with two phytoplankton types (general phytoplankton and diazotrophs), one zooplankton type, and chemical tracers nitrate, phosphate, dissolved inorganic carbon, alkalinity, and oxygen. Iron limitation is accounted for using a seasonally variable mask of dissolved iron concentrations in the upper three ocean layers (Keller et al., 2012). The biogeochemistry is comprehensively assessed in Keller et al. (2012), however, the model has since been updated with several bug fixes and minor

adjustments. Only one of the bug fixes is relevant to our study. In previous published versions of the model, the depth was incorrectly calculated for the light availability equation in a way that resulted in too much light in the first ocean depth level. This calculation is corrected here.

Our study examines model biogeochemical sensitivity to a spread of $k_c$ values at both equilibrium in a pre-industrial climate (atmospheric $CO_2$ concentration of 283.8 ppm) and a future projection. We use historical atmospheric $CO_2$ concentrations,

agricultural land cover, volcanic radiative forcing, sulphate aerosol and CFC concentrations to force the model, as well as changes in land ice and solar forcing from year 1800 to 2005 following Machida et al. (1995); Battle et al. (1996); Etheridge et al. (1996, 1998); Flückiger et al. (1999, 2004); Ferretti et al. (2005); Meure et al. (2006). From year 2005 to 2300 the simulations were forced using increasing $CO_2$ and non-$CO_2$ greenhouse gas concentrations, projected changes to the fraction of the land surface devoted to agricultural uses (calculated to year 2100 by Hurtt et al., 2011, and then held constant after),

and changes in the direct effect of sulphate aerosols following "business-as-usual" RCP scenario 8.5 (RCP8.5, Riahi et al., 2007; Meinshausen et al., 2011). Solar insolation at the top of the atmosphere, wind stress, and wind fields varied seasonally (Kalnay et al., 1996), and the wind fields were geostrophically adjusted to air temperature anomalies (Weaver et al., 2001). The sediment and weathering models (Meissner et al., 2012) were not used. Model equilibration was achieved by integrating over 10,000 years prior to application of climate forcing.

The simplistic nature of Equation 1 makes our study highly idealized. Parameter $k_w$ represents light attenuation of water and is fairly well-constrained to about $0.04 \, \text{m}^{-1}$ (Lorenzen, 1972), which is its assigned value in our model. The light attenuation of ice parameter is not examined here: any primary production sensitivity to variation in $k_i$ is likely to have effects relegated to the high latitudes. Light attenuation by phytoplankton also implicitly accounts for attenuation of light by co-varying factors, with the current default model value applied to both diazotrophs and the single general phytoplankton type (Eqn. 1). The

Schmittner et al. (2008) $k_c$ value of $0.03 \, \text{(m mmol N m}^{-3})^{-1}$ was increased in Keller et al. (2012) to $0.047 \, \text{(m mmol N m}^{-3})^{-1}$. Light attenuation parameters are measured based on chlorophyll (commonly Chlorophyll $a$) concentration but the model uses nitrogen units, necessitating the application of a conversion factor also implicit to $k_c$. Conversion of the range of $k_c$ values given above ($0.006$ to $0.041 \, \text{m}^2(\text{mg Chl } a)^{-1}$) to carbon and then nitrogen units using the maximum Chl $a$ to carbon ratio for non-diatom phytoplankton in Table 4 of Dutkiewicz et al. (2015) and the Redfield C:N ratio used in our model (6.625)

yields a range of $0.008$ to $0.054 \, \text{(m mmol N m}^{-3})^{-1}$ in the observationally-based literature (though higher values in models exist- Evans and Parslow 1985 used a value of $0.12 \, \text{(m mmol N m}^{-3})^{-1}$). For our test, we employ eight separate simulations using $k_c = 0.01, 0.02, 0.03, 0.04, 0.05, 0.06, 0.07$, and $0.08 \, \text{(m mmol N m}^{-3})^{-1}$. Increasing the light attenuation parameter value increases the self-shading effect of the phytoplankton biomass, reducing the amount of light available for photosynthesis. In the following analysis they will be referred to as 'K1-8', as we assess the impact of parameter choice on model net primary

production, carbon and nutrient distributions in a model equilibrated to pre-industrial climate conditions and then forced with

historical and projected greenhouse gas concentrations. Gridded observations from the World Ocean Atlas climatology (Garcia et al., 2010a, b) (phosphate, nitrate, and oxygen) and GLODAP (Key et al., 2004) (DIC and alkalinity) are compared to the pre-industrial model.

## 3 Results

### 3.1 Pre-industrial equilibrium simulations

Patterns of equilibrated net primary production (NPP) in the UVic ESCM are sensitive to $k_c$ within the tested range. Depth-integrated zonally and annually averaged NPP, and horizontally and annually averaged NPP are shown in Figure 1. The simulation spread is smallest between 20° and 40°, where phytoplankton biomass is low. The tropics and regions between 40° and 75°, particularly in the Southern Ocean, show the largest differences in NPP with varying $k_c$. In the tropics, the differences in simulated zonal mean NPP between end members K1 and K8 exceeds 200 g C $\text{m}^{-3}\text{y}^{-1}$, while regional differences in depth-integrated NPP exceed 200 g C $\text{m}^{-3}\text{y}^{-1}$ over large parts of the Southern Ocean and tropics (Fig. 2). In the Southern Ocean, K1 zonally averaged primary production rates can exceed those of K8 by more than a factor of 3 because phytoplankton in K1 do not self-shade as strongly during the Austral summer, thereby allowing for a stronger seasonal cycle. South of this region (around 60°S) UVic ESCM primary production transitions to being light-limited from being nutrient-limited to the north (annually averaged limitation regimes are shown in Fig. 3) and so reducing the self-shading increases primary production in the light-limited regime. The transition zone between light and nutrient limitation is well-mixed, and lateral advection of regenerated nutrients from the light-limited regime boosts NPP in the nutrient-limited regime in low-$k_c$ value simulations. In the more stratified (and nutrient-limited) tropics, the effect is opposite in that K8 yields zonally averaged NPP of up to double K1 because stronger self-shading inhibits deeper photosynthesis (see the globally averaged NPP depth profile plot in Fig. 1, which is dominated by the low latitude response), making more regenerated nutrients available at the surface (Figs. 2 and 4, and similar to the effect of light attenuation by CDOM described previously by Kim et al. 2015). Higher nutrient concentrations at the tropical surface in K8 cause a net increase in depth-integrated primary production because of the temperature dependency of primary production and respiration in the model (the warmer surface increases the production and remineralisation rates, resulting in higher NPP). Simulation differences in the tropical eastern Pacific upwelling region arise from processes similar to those described in the Southern Ocean. While the eastern Pacific upwelling zone is nutrient-limited in our model (like the rest of the tropics, Fig.3), a weak near-surface temperature gradient reduces primary production in the surface layer. Higher light availability in K1 therefore allows for deeper utilization of upwelled nutrients, resulting in higher depth-integrated NPP in K1 compared to K8. Three distinct regional responses to $k_c$ parameter value choice are therefore apparent. In regions that are light-limited, reducing the light attenuation parameter results in higher NPP (Southern Ocean and Arctic). In regions that are nutrient-limited, reducing the light attenuation parameter results in lower NPP when combined with a strong vertical temperature gradient near the surface (tropics and subtropical gyres). In regions that are nutrient-limited and are characterized by a weak vertical temperature gradient near the surface, reducing the light attenuation parameter results in higher NPP (eastern Pacific, western boundary currents). The northern hemisphere midlatitudes do not show as clear a zonally-averaged NPP-$k_c$

relationship as can be seen in the southern hemisphere and the tropics because the western boundary currents and oligotrophic regions oppose each other in the north Atlantic and north Pacific.

Carbon and nutrient distributions in the UVic ESCM are also sensitive to $k_c$ because parameter choice affects the efficiency of the biological pump (Fig. 1), leading to a redistribution of nutrients (Fig. 4). Low-value $k_c$ simulations experience a greater proportion of global NPP in the high latitudes (regions with higher sequestration efficiency; DeVries et al. 2012), and increasing the $k_c$ value shifts NPP towards the tropics (a region of lower sequestration efficiency; DeVries et al. 2012). As a consequence, more nutrients and carbon end up in the abyssal Pacific Ocean in low-value $k_c$ simulations than in higher value ones. Increased storage of nutrients in this deep ocean basin reduces the inventory available for subduction in the northern Atlantic (e.g., Kwon and Primeau, 2006; Kwon et al., 2009; Kriest et al., 2012), where water column concentrations of nitrate and phosphate decline (Fig. 4). Increasing $k_c$ values reduces average surface alkalinity (Fig. 4) by about 50 $\mu$mol kg$^{-1}$ globally, a response to increasing low latitude NPP (including a stronger carbonate pump) with higher $k_c$ values. In low-value $k_c$ simulations, alkalinity is higher in the Atlantic as a result of the decline of the Atlantic biological pump. Deep ocean alkalinity is less sensitive to $k_c$ value, though the average deep Pacific also shows a range of about 50 $\mu$mol kg$^{-1}$ and the Southern Ocean varies by about 25 $\mu$mol kg$^{-1}$. Deep ocean DIC, however, is more sensitive to choice of $k_c$ value (Fig. 4). K4-K8 DIC range in basin averages less than 30 $\mu$mol kg$^{-1}$ but sensitivity increases at lower $k_c$ values. K1 deep DIC values are about 40 $\mu$mol kg$^{-1}$ higher in the global average, Pacific, and Southern Ocean basins than K2. These higher deep DIC values are a consequence of higher NPP in the high latitudes owing to a weaker self-shading effect, which increase carbon and nutrient export to the deep ocean. Phosphate and nitrate basin-averaged profiles show a range of values generally proportionate to the range in DIC, with drivers of the differences being the same (lower $k_c$ values yield higher global NPP, lower surface nutrients, and higher deep ocean nutrients, as well as a shift in NPP to higher latitudes, Fig. 4). Likewise, deep ocean oxygen is lower for lower $k_c$ values because there is more deep ocean remineralisation (Fig. 4). The global average deep ocean oxygen concentration has a range of about 100 mmol m$^{-3}$, which is about half of the average deep ocean content. The Southern Ocean and Pacific show similar oxygen sensitivity.

Which $k_c$ value performs the "best" with respect to biogeochemical observations is not thoroughly quantified here, but generally K4 and above perform better with respect to deep ocean nutrients and oxygen, K2 to K5 do better with respect to global DIC, and K1 and K2 outperform the others with respect to global alkalinity (see global RMSE values in Fig. 4). As discussed in Section 1, selection of a single model $k_c$ value to represent all ocean biology and co-varying factors is fairly ad-hoc, and consequentially, the range of values we selected for this sensitivity comparison are as well. Based on Figures 1 and 4, the two lowest $k_c$ values we selected perform anomalously with respect to the others (higher Southern Ocean NPP, deep ocean DIC, nitrate, and phosphate and much lower deep ocean oxygen). Therefore we will examine two groups of K values in Section 3.2, K1-K8 (the full range tested) and K3-K8 (the subset better reproducing modern deep ocean biogeochemistry).

## 3.2 RCP 8.5 transient simulations

Figure 5 plots the increase in atmospheric CO$_2$ concentration from 283.8 ppm to 1962 ppm over the course of the transient integration. The physical response is the same across all simulations and closely follows that described in Kvale et al. (2015).

Zonally averaged ocean surface temperatures rise by as much as 10°C, North Atlantic maximum meridional overturning reduces from 20 to 9 Sverdrups (not shown), and widespread near-surface stratification occurs (Fig. 5). The phytoplankton and zooplankton respond to surface warming by increasing metabolic rates, and microbial fast recycling in the near-surface increases (Kvale et al., 2015). Stratification reduces the availability of nutrients in the near-surface. The global response in NPP

until about the year 2100 depends on the simulation, with K1-K6 showing a decline, and K7 and K8 showing no change and an increase in NPP, respectively. After about year 2100, global NPP in K4-K8 increases linearly, while global NPP increases at a declining rate in K1-K3.

Model spread in global NPP response generally increases with radiative forcing. Change in global NPP differs by 2.5 Pg C $y^{-1}$ by 2100 (more than 100% of the total change in NPP at 2100 for all simulations) and $k_c$ parameter choice can determine

the sign of the change. This applies even if only considering the subset of $k_c$ parameters offering the better fits to pre-industrial nutrient and carbon observations (K3-K8) and excluding K1 and K2. By 2300 this spread has increased to 7 Pg C $y^{-1}$ across all simulations, and 5 Pg C $y^{-1}$ between K3 and K8. By 2300, the spread between K3 and K8 is roughly equal to the all-simulation average change in NPP since 1800, suggesting choice of $k_c$ value can have a significant and increasing effect on global NPP response to climate forcing.

Before the year 2100, physical limitation of nutrients is the dominant driver of changes in global NPP (Kvale et al., 2015). Over this time, choice of $k_c$ parameter value affects the magnitude and direction of how NPP in different regions responds. Increasingly oligotrophic conditions expand the extent of nutrient-limited regions, with phosphate limitation arising in the tropics in lower $k_c$ value simulations (shown for K1 and K3 in Fig. 3). More nutrient limitation results in declining global NPP in the simulations with weaker self-shading (K1-K6). In these simulations, lower starting concentrations of surface nutrients

causes the biology in these simulations to be more sensitive to an increase in stratification. Figure 6 (left plot) shows some decline between years 1800 and 2100 in tropical depth-integrated NPP between 10° and 20° for all simulations, with the declines generally increasing with decreasing $k_c$. Declines in low latitude NPP in simulations K1-K6 are not fully offset by increasing NPP in the Southern Ocean, which is driven by regional increasing temperature, wind-driven overturning, and nutrient remineralisation (Kvale et al., 2015). K1 and K2 demonstrate a particularly strong increase in NPP in the Southern

Ocean around 60°S, for reasons discussed below. Along the Antarctic margin (around 80°S), local freshening causes large local declines in NPP in simulations using weaker self-shading, though the region is not nutrient-limited in our model. The mechanism for the decline is a drop of seawater temperature in the second ocean depth layer, which disproportionately affects simulations that have deeper NPP. Simulations K7 and K8 are relatively less sensitive to increasing stratification (and associated nutrient limitation) because their high $k_c$ values raise primary production higher in the water column, thereby raising surface

nutrient concentrations and allowing the phytoplankton to be less reliant on resupply of nutrients from deeper waters. Pre-2100 global NPP increases in K8 are therefore attributable to increased biological rates due to warming. All simulations show an increase in NPP north of about 50°N to 60°N, which is driven by warming temperatures in all light attenuation tests.

After year 2100, physical limitation of nutrients becomes a less important driver of changes in global NPP than temperature-enhanced biological processes (Kvale et al., 2015). Increasing global NPP in lower $k_c$ simulations is dominated by increasing

NPP in the Southern Ocean, and in higher $k_c$ simulations is a combination of increasing NPP in the Southern Ocean and low

latitudes. The drivers of change in NPP after year 2100 in the Southern Ocean are the same as those mentioned earlier, with alleviation of light limitation (Fig. 3) and warming seawater temperatures increasing production rates, to particular effect on low-value $k_c$ simulations. The driver of change in NPP after year 2100 in the tropics is the increase in temperature-enhanced biological processes. Increasing divergence in zonal mean NPP between simulations is shown in Figure 6 (right plot). At year

1800, Southern Ocean depth-integrated and zonally-averaged NPP in K1 exceeds that in K8 by 180 g C m$^{-3}$y$^{-1}$ at most. By year 2100, this difference has increased to 250 g C m$^{-3}$y$^{-1}$, and by year 2300 it is over 300 g C m$^{-3}$y$^{-1}$ (corresponding to more than 150% of K1 pre-industrial Southern Ocean zonal mean NPP). The pattern is similar but opposite in the low latitudes, where depth-integrated and zonally-averaged NPP in K8 exceeds that in K1 by about 200 g C m$^{-3}$y$^{-1}$ in year 1800, around 250 g C m$^{-3}$y$^{-1}$ in year 2100, and more than 300 g C m$^{-3}$y$^{-1}$ in year 2300. Divergence in depth-integrated and zonally

averaged NPP for K3 and K8 follows the same pattern with smaller magnitudes- K3 exceeds K8 in the Southern Ocean by as much as 75 g C m$^{-3}$y$^{-1}$ (year 1800) and 100 g C m$^{-3}$y$^{-1}$ (year 2300, a difference of 60% of the highest pre-industrial K3 Southern Ocean zonal NPP value), and K8 exceeds K3 in the low latitudes by up to about 130 g C m$^{-3}$y$^{-1}$ (year 1800) and over 200 g C m$^{-3}$y$^{-1}$ (year 2300). Throughout the simulations, northern hemisphere differences between K1 and K8 and K3 and K8 are relatively small, as the regional trends in responses to nutrient limitation are less cohesive than in the other regions.

The biogeochemical consequences of $k_c$ parameter choice at years 1800 and 2300 are shown for major ocean basins in Figure 7. Most biogeochemical quantities retain the pre-industrial spread in global profiles with increasing $CO_2$ forcing (for both K1-K8 and K3-K8), with changes on basin-scale canceling out in the global mean. This observation likely relates to the asymmetry in Southern Ocean/tropical trends in NPP K1-K8 differences noted in the previous paragraph (NPP rates in K1 exceed those of K8 in the Southern Ocean roughly equally to the amount NPP rates in K8 exceed those of K1 in the tropics). Basins reveal

increasing changes with time. In particular, biogeochemical quantities in the deep Southern Ocean display increasing sensitivity to light attenuation parameter choice with time. At 5000 meters depth, by year 2300 the K1-K8 difference in average Southern Ocean alkalinity is 50 $\mu$mol kg$^{-1}$ (compared to 25 $\mu$mol kg$^{-1}$ at year 1800), while the difference in DIC has increased to 170 $\mu$mol kg$^{-1}$ (110 $\mu$mol kg$^{-1}$ at year 1800). Alkalinity and DIC are higher in the K1 simulation because of higher NPP and stronger associated carbon export to the deep Southern Ocean compared to K8. Phosphate concentration differences at year

2300 and 5000 m depth equal 0.8 mmol m$^{-3}$ (0.5 mmol m$^{-3}$ at year 1800), nitrate concentration differences exceed 10 mmol m$^{-3}$ at year 2300 (9 mmol m$^{-3}$ at year 1800), and oxygen concentration differences equal 180 mmol m$^{-3}$ at year 2300 (120 mmol m$^{-3}$ at year 1800).

All simulations experience a loss in oxygen due to warming and increasing remineralisation, but K1 and K2 additionally experience denitrification in the Southern Ocean (not shown) as a result of very high primary production in the region and

already lower oxygen concentrations at pre-industrial equilibrium. This denitrification establishes a nutrient feedback with the low latitude Pacific and Indian Oceans that reduces Southern Ocean oxygen further (Fig. 8), thus producing a strong regional decline in oxygen despite K1 and K2 showing weaker global NPP trends than the other simulations. The feedback starts with increased stratification in the low-latitude Pacific and Indian Oceans (to which low $k_c$ simulations are particularly sensitive), which limits nitrate availability for local primary production. As a result, more phosphate begins to advect into the Southern

Ocean, where it fertilizes phytoplankton growth. The regional loss of phosphate in the western Pacific and Indian Oceans

in K1 and K3 can be seen in Figure 3 as expanding phosphate-limited regimes. Warming seawater increases both primary production and remineralisation rates. Phytoplankton in K1 and K2 are only weakly inhibited by self-shading and take full advantage of warmer temperatures and imported nutrients, and the resulting large increases in primary production leads to the consumption of enough oxygen that denitrification establishes in the Southern Ocean. Denitrification reduces the flow of

nitrate in intermediate water back into the low latitude Pacific and Indian basins, which become even more nitrate-limited.

Excluding the K1 and K2 simulations in which Southern Ocean denitrification occurs, differences between K3 and K8 biogeochemical quantities change less over time, though the spread in Southern Ocean intermediate and deep nitrate concentrations increases by about 50% (Fig. 7).

## 4 Discussion

The pre-industrial equilibrium simulations demonstrate a sensitivity in zonally averaged NPP, and global and basin profiles of biogeochemistry to choice of $k_c$ value for the range tested. Simulation spread is greatest in the Southern Ocean and tropics. Simulation differences arise from a complex interplay of factors. Higher light attenuation values have a stronger self-shading effect on phytoplankton, which acts to increase NPP in stratified, nutrient-limited regions (the low latitudes in our model). This is because strong self-shading raises the depth profile of primary production into the warmest surface ocean layer by

reducing the amount of light available in lower layers. Biological processes have a temperature dependency in the UVic ESCM, so reduced nutrient utilization in deeper layers increases nutrient availability at the surface, increasing surface net primary production. This finding somewhat agrees with Kim et al. (2015) who found a decoupling between nutrient concentrations and biomass when light attenuation of CDOM was accounted for in their ESM. Including light attenuation of CDOM (therefore raising total model light attenuation) increased surface nutrient concentrations in their model through a similar mechanism

(shoaling of the biomass and production profiles), however they found CDOM light attenuation decreased depth-integrated biomass and attributed the increasing surface nutrients to less total production. Our model demonstrates an increase in depth-integrated NPP with increasing light attenuation. The Kim et al. (2015) model allowed biological light attenuation to reduce shortwave heating of the water column while our model does not account for this. Including a reduction of near-surface temperatures with strong self-shading might reduce the increase we find in NPP with higher values of $k_c$, though Manizza et al.

(2005) found inclusion of a shortwave feedback to NPP can also enhance spring sea surface temperatures and reduce sea ice.

Our model is also nutrient-limited in the tropical eastern Pacific upwelling zone, but this region is also well-mixed (near-surface shoaling of the nutrient profile is weaker and the near-surface temperature gradient is lower) so greater light inhibition of deeper photosynthesis results in less NPP with higher $k_c$ values. In the Southern Ocean, higher NPP is the result of lower light attenuation values, which allow phytoplankton in the light-limited regions to produce deeper in the water column. In

simulations with lower light attenuation values, NPP also increases in the northern parts of the Southern Ocean, which are nutrient-limited in our model. This may be a response to nutrient advection from locally increased near-surface remineralisation arising from higher NPP in the light-limited regions, and highlights the important point that light attenuation parameter choice can potentially have surprising effects on nutrient transport by changing the depth of primary production. Keller et al. (2016)

found a similar effect at this latitude by suppressing primary production around Antarctica, which caused a northward advection of nutrients, raising local NPP. In this particular region, higher vertical resolution might reduce the overall NPP response of the Southern Ocean to decreasing light attenuation parameter by reducing advected regenerated nutrients and reducing preformed nutrients made available for primary production by reduced self-shading. In the stratified low latitudes, higher vertical resolution might reduce the nutrient shoaling effect of strong self-shading.

Though iron availability is accounted for in the form of a seasonally-variable mask, in our model iron is not a limiting nutrient on an annually-averaged basis. This is in contrast to evidence of iron limitation in the Southern Ocean, North Atlantic, and eastern boundary currents and upwelling systems (see recent review by Tagliabue et al., 2017). More iron limitation of phytoplankton growth in the UVic ESCM might damp the NPP response we show for lower light attenuation simulations in the Southern Ocean and eastern equatorial Pacific. More iron limitation might also mitigate differences in the efficiency of the global biological pump between high and low-value light attenuation parameter simulations. Higher NPP in the high latitudes in low-value light attenuation parameter simulations results in more efficient export and storage of nutrients in the deep ocean, particularly the abyssal north Pacific (also found by DeVries et al. 2012). Model phosphate is conserved in our simulations, thus larger deep ocean inventories result in lower concentrations in downstream surface and intermediate waters (in qualitative agreement with Kwon and Primeau 2006; Kriest et al. 2012). The effect of enhanced deep nutrient sequestration is most apparent in Atlantic phosphate and nitrate profiles, where concentrations are lower for lower $k_c$ simulations and NPP is not very much higher at the surface, in spite of being a seasonally well-mixed region. If iron was more limiting in the Southern Ocean deep water formation regions, fewer nutrients would be sequestered in the deep Pacific and more would be available to the north Atlantic, raising regional primary production and export (assuming no iron limitation also existed in the north Atlantic). More iron limitation in the low latitudes might furthermore damp the NPP response of higher $k_c$ simulations in the thermally stratified tropics, thus increasing nutrient transport poleward and increasing high latitude NPP.

Parameter estimation and the quantification of biogeochemical model uncertainty is a major field of research (see review by Schartau et al., 2017). Our study demonstrates the importance of considering transient model behaviour both in parameter estimation and estimates of uncertainties for biogeochemistry in OGCMs and ESMs. Differences in the relative importance of regional biological pumps to global NPP between high and low light attenuation simulations has a strong effect on how the model responds to climate change. Using a lower $k_c$ value emphasizes the Southern Ocean response, where physical drivers of warming temperatures and increasing light availability enhance NPP, while using higher $k_c$ values place increasing importance on the tropical drivers of warming and stratification. Thus, simulation spread increases in our transient simulations, and $k_c$ parameter choice can determine the sign as well as the magnitude of the global NPP response, particularly in the near-term (to year 2100), when physical changes are the dominant model drivers of NPP. Simulations K6 to K8 perform roughly equivalently with respect to biogeochemical observations in the pre-industrial equilibrium, and yet global NPP differences by 2100 are around 2 Pg C y$^{-1}$, with K8 showing an increase and K6 showing a decrease with respect to year 1800 NPP. A recent review of the drivers of change in global NPP in a suite of OGCMs and ESMs to which climate forcing was applied found the low latitudes contained the largest spread in model response, with global trends comprising a balance between increasing metabolic

rates and increasing stratification (Laufkötter et al., 2015). Our results suggest differences between this balance across models might be partly related to differences in the treatment of phytoplankton light attenuation.

Climate change furthermore compounds regional differences in biological pumps according to light attenuation parameter choice, as nutrient export from the tropics to the Southern Ocean increases in low-$k_c$ value simulations due to the disproportionate sensitivity of low-$k_c$ value simulations to low latitude stratification. Low-latitude nutrient recycling is strongly enhanced in high-$k_c$ value simulations due to surface warming and shallow NPP, reducing the availability of regenerated nutrients for export to the high latitudes and damping increasing NPP in the Southern Ocean. Where NPP responds to climate change has implications for long-term carbon sequestration, which can be seen as a 180 $\mu$mol kg$^{-1}$ deep Southern Ocean DIC surplus and rising DIC concentrations in the near-surface Atlantic in the K1 simulation when compared to K8 by year 2300. Furthermore, ocean oxygen shows particular sensitivity to light attenuation parameter choice when forcing the model with future projections. That oxygen is sensitive to model treatment of NPP (e.g., Kriest et al., 2012), and that Southern Ocean biological processes can affect global nutrient, carbon, and oxygen distributions (e.g., Kwon and Primeau, 2006; DeVries et al., 2012; Kriest et al., 2012; Keller et al., 2016) are not new findings but, as far as we know, our study is the first to demonstrate the potential for denitrification in the Southern Ocean. The nutrient exchange feedback that establishes in the two lowest $k_c$ value simulations K1 and K2 substantially reduces Southern Ocean oxygen concentrations. Our model is highly idealized but it is worth noting that the nutrient exchange feedback occurs because: 1) it highlights the potential for strong biogeochemical teleconnection between the Southern Ocean and the low latitude Pacific in the real world, and 2) light attenuation characteristics of dominant phytoplankton (Katz et al., 2004) and ocean oxygen content (Lenton et al., 2014) and rates of change (e.g., Paleocene Eocene Thermal Maximum; Norris et al., 2013) have changed over geologic timescales. A recent model study by Meyer et al. (2016) explored the sensitivity of oxygen to e-folding depth of remineralisation and total phosphate inventory and hypothesized an increase in remineralisation depth has occurred over the Phanerozoic alongside a stabilisation of ocean oxygen inventory. Our tests demonstrate another potential mechanism for the increase in ocean oxygen inventory in equilibrated conditions as well as for a stabilisation of oxygen under rapid climate change- an evolved increase in light attenuation by dominant phytoplankton, which in our model increases ocean oxygen inventory and mitigates total oxygen change with climate forcing.

It is possible that primary production in our model demonstrates similarly increasing sensitivity to other phytoplankton parameters with climate change, and that the sensitivity of NPP to $k_c$ may be damped or magnified by the choice of other parameter values (e.g., the initial value of the photosynthesis-irradiance curve). Exploring the uncertainty associated with multiple parameter manipulations is costly and better left to offline approaches that can objectively and systematically assess the solution space (see review by Schartau et al. 2017), though as far as we know, offline methods for 3-D models are currently restricted to steady-state analysis. It is also possible that including a fully resolved radiative transfer model and explicit IOPs for multiple phytoplankton types could damp the Southern Ocean response we find in K1 and K2 and the low latitude response we find in the higher light attenuation simulations (Gregg and Rousseaux, 2016). Lastly, the impact of phytoplankton shading on water column heating is not considered here. This is a potentially significant omission with respect to the climate change response of model physics as global net primary production increases strongly in all of our simulations but never contributes to regional cooling, in contrast to the Manizza et al. (2005) finding that light attenuation by biomass can amplify the seasonal

cycle of temperature, mixed layer depth and ice cover by about 10% under pre-industrial conditions. From a global perspective, increasing shortwave penetration along the equator can warm regions to the south (Gnanadesikan and Anderson, 2009), which might damp southward nutrient transport in our low-light attenuation simulations by increasing local export production. However, increasing shortwave penetration in the Southern Ocean can enhance mode water formation from subtropical water (Gnanadesikan and Anderson, 2009), which might enhance the positive nutrient feedback we demonstrate in low-$k_c$ simulations. Regardless, the UVic ESCM is fairly typical with respect to other ESMs in regard to the treatment of the underwater light field, therefore our sensitivity study is useful for assessing uncertainty in models of similar structure.

## 5 Conclusions

The highly simplistic parameterisation of underwater light availability used in the UVic ESCM to calculate primary production and associated chemistry (alkalinity, DIC, nitrate, phosphate, and oxygen) is sensitive to a range of light attenuation parameter values constrained by data. This applies both to pre-industrial equilibrium and future projections. This sensitivity can grow with changing background climate as complex biogeochemical feedbacks develop, with primary production and ocean oxygen being especially susceptible to parameter choice. Our study highlights the need to assess biogeochemical models under transient as well as equilibrium conditions. In addition, the biogeochemical feedback we describe in two of our transient simulations also serves as a reminder that even seemingly small events, like the emergence of shell-secreting phytoplankton, could have potentially large biogeochemical consequences just by altering the underwater light field.

## 6 Code availability

Model data and figure scripts are available from https://thredds.geomar.de/thredds/catalog/open_access/kvale_meissner_2017_bg/catalog.html. Model code is available from the authors upon request.

*Author contributions.* K. Kvale designed and implemented the experiment, and wrote the paper with comments by K. Meissner. K. Kvale and K. Meissner interpreted the model results.

*Acknowledgements.* Computing resources were provided by Kiel University.

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

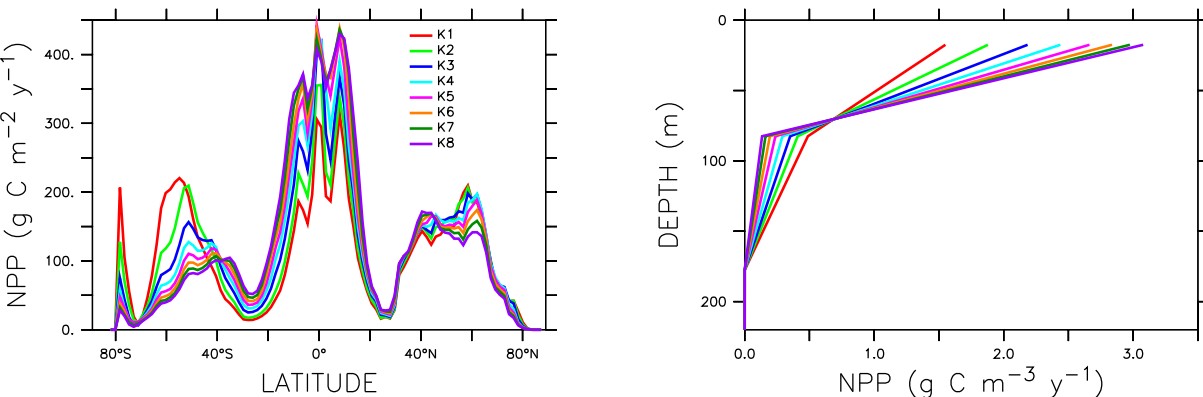

**Figure 1.** Annually and zonally averaged pre-industrial, depth-integrated NPP (left), and annually and globally averaged NPP with depth (right).

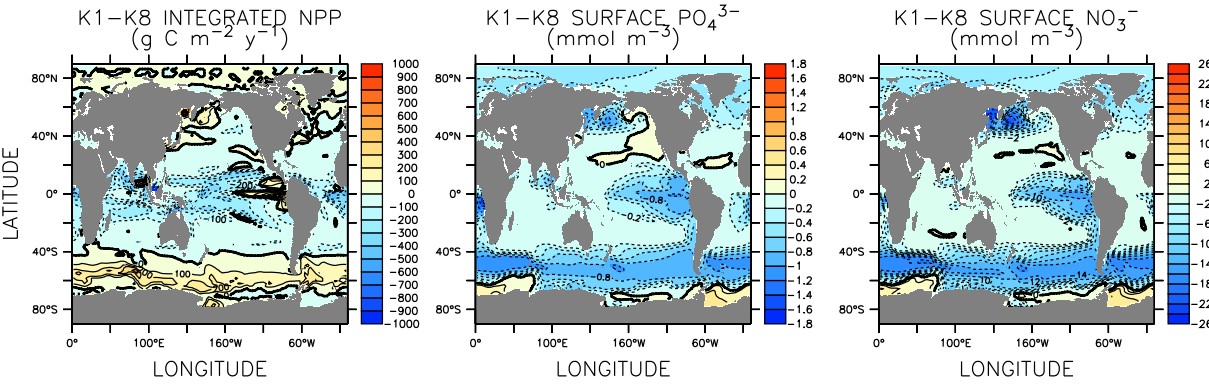

**Figure 2.** Annually averaged pre-industrial differences between K1 and K8 for depth-integrated NPP (left), surface phosphate (middle) and surface nitrate (right).

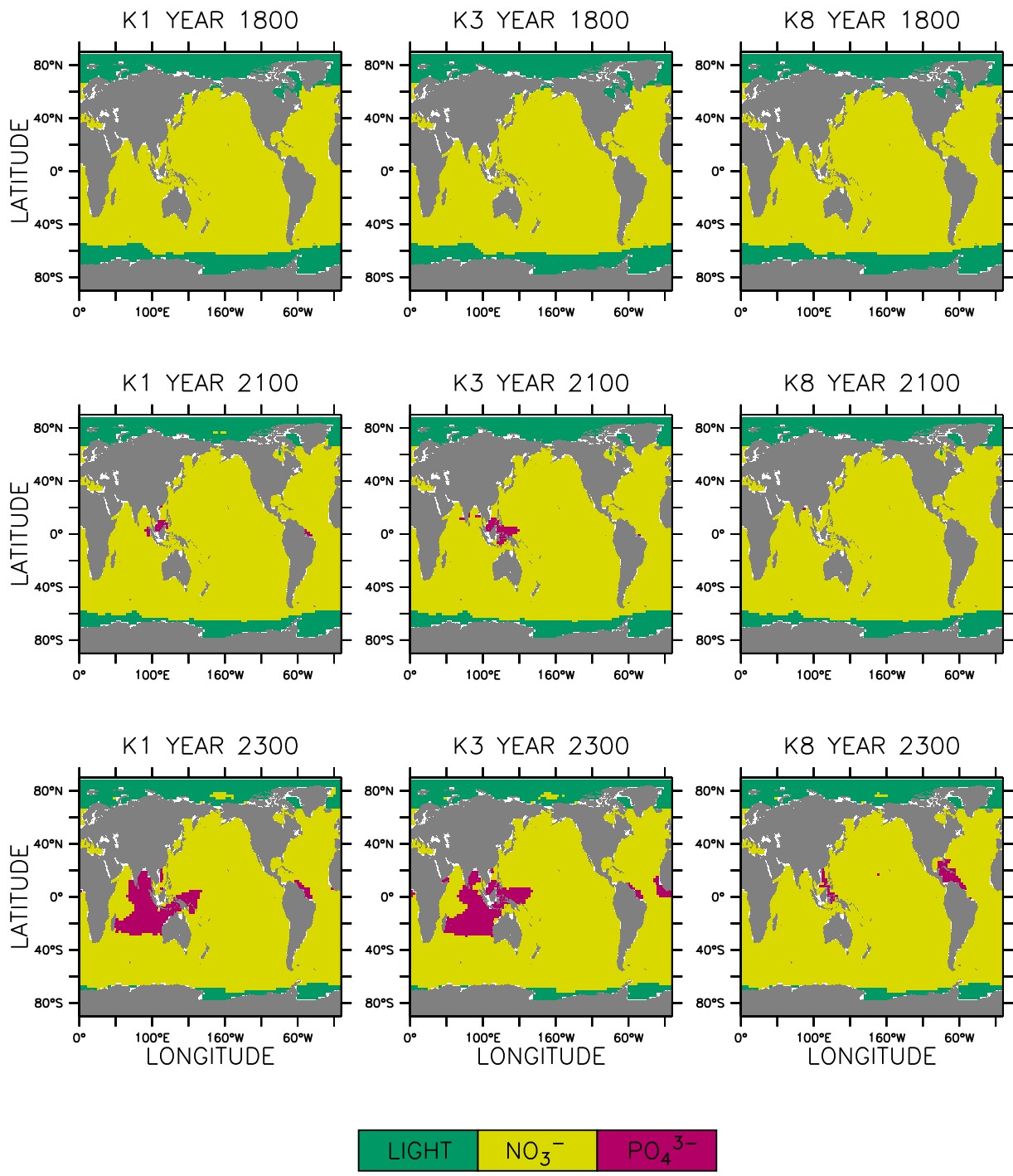

**Figure 3.** K1, K3, and K8 light and nutrient limitation regions for years 1800, 2100, and 2300.

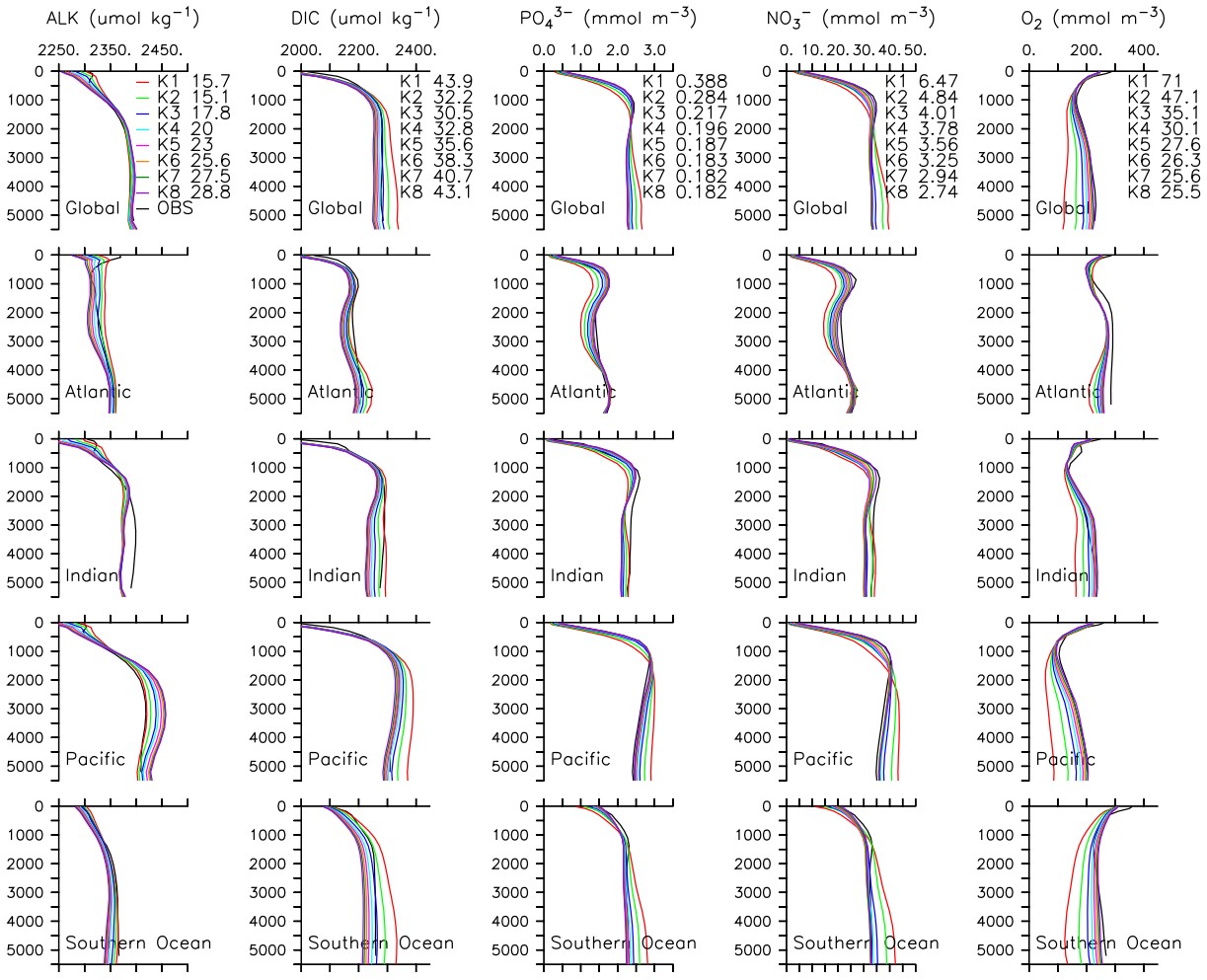

**Figure 4.** Annual mean pre-industrial biogeochemical tracer profiles averaged by ocean basin for all simulations compared to gridded observations. Global root-mean-square-error is provided for each simulation.

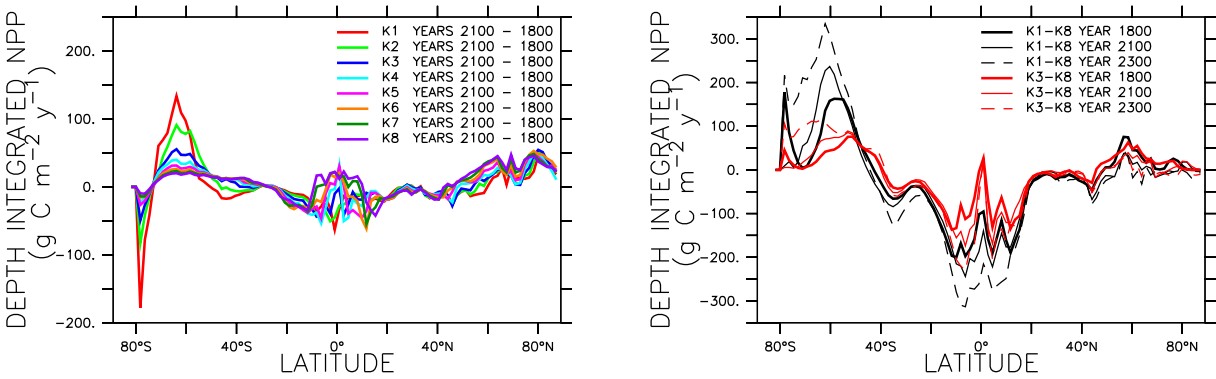

**Figure 5.** Atmospheric $CO_2$ concentration forcing of all simulations (top left). Zonally averaged change in sea surface temperature in all simulations (top right). Change in global average density gradient with depth in all simulations (bottom left). Change in globally integrated net primary production in all simulations (bottom right).

**Figure 6.** Change in annually and zonally averaged, depth integrated NPP between years 2100 and 1800 (left plot) and annually and zonally averaged, depth integrated NPP differences between K1 and K8 (black lines) and K3 and K8 (red lines) at three time slices (right plot).

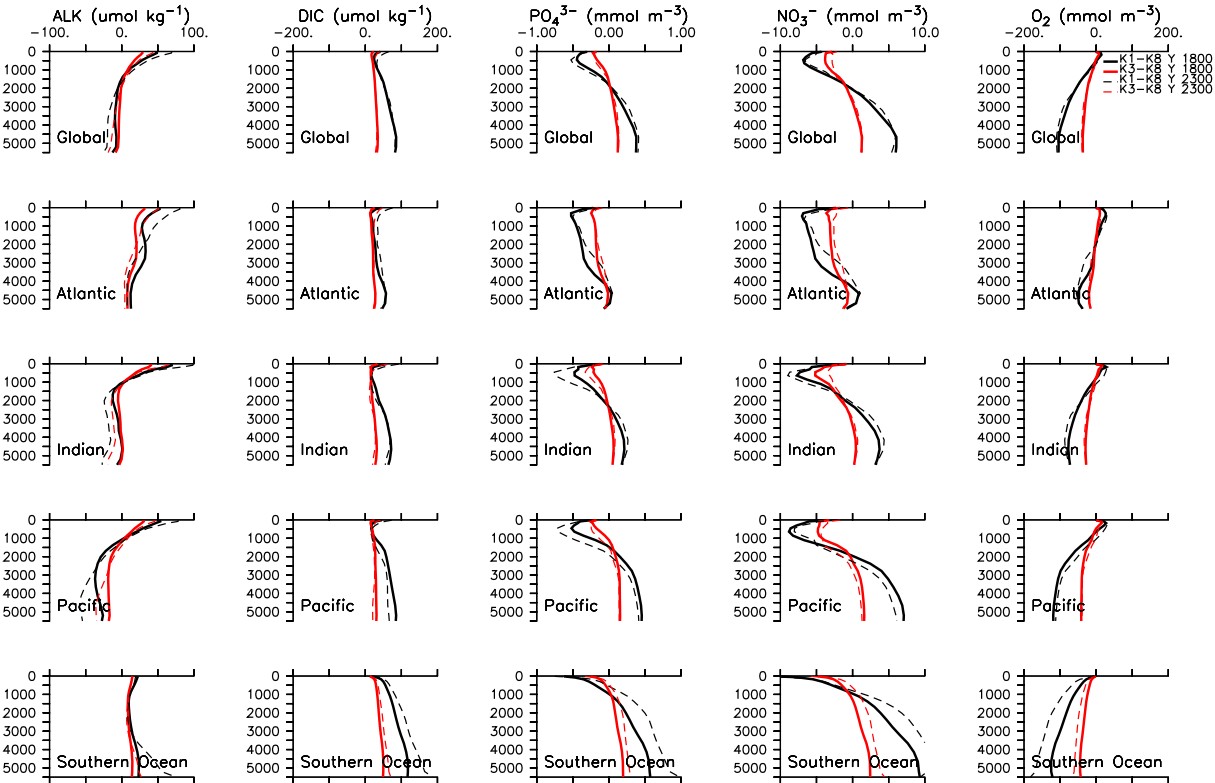

**Figure 7.** Annual mean biogeochemical tracer profile differences between K1 and K8 (black lines) and K3 and K8 (red lines), averaged by ocean basin for all simulations at years 1800 (solid lines) and 2300 (dashed lines).

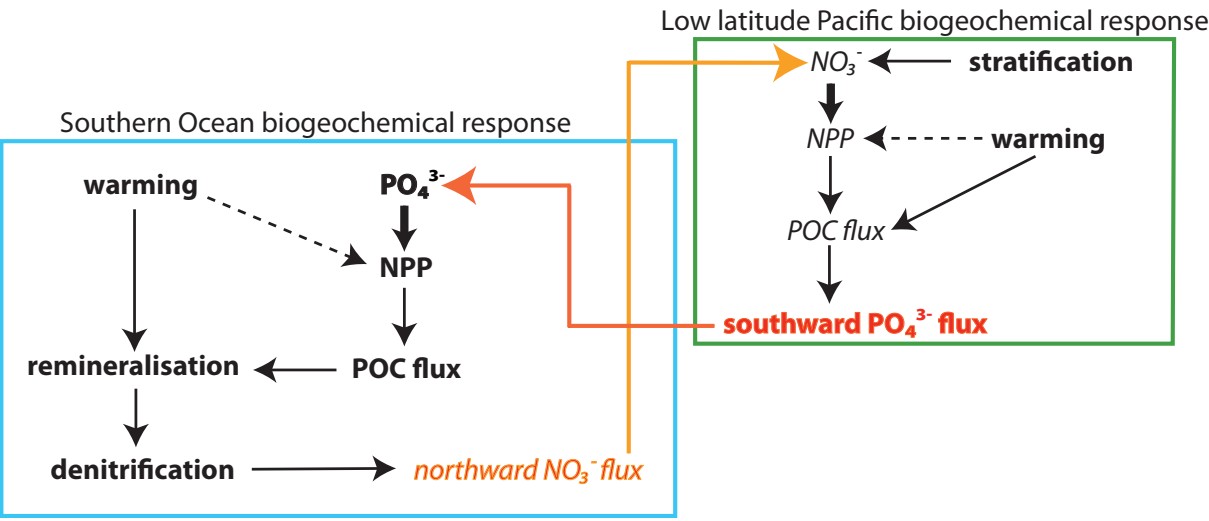

**Figure 8.** K1 and K2 feedback schematic in Southern Ocean and low latitude Pacific nutrients. Increases with climate change are represented in bold font. Decreases with climate change are represented in italic font. Regular font indicates little or no change with climate forcing. Bold arrows indicate the dominant factor influencing change in NPP. Dashed arrows indicate the secondary factor influencing change in NPP. Nutrient feedback between regions is shown in coloured arrows.