# Peer review of "Primary production sensitivity to phytoplankton light attenuation parameter increases with transient forcing"

_Biogeosciences, 2017_

## Referee Comment (RC1) · Anonymous Referee #1 · 9 May 2017

General comments:

The paper "Primary production sensitivity to phytoplankton light attenuation parameter increases with transient forcing" by Kvale and Meissner addresses a topic that is both relevant and within the scope of BG, namely, the sensitivity of simulated marine biogeochemistry to changes in the parametrization of the under water light field under climate change. In particular, the study addresses the sensitivity of marine net primary production (NPP), alkalinity, dissolved inorganic carbon, phosphate, nitrate, and oxygen, as modelled with the University of Victoria Earth System Climate Model (UVIC ESCM) in pre-industrial equilibrium as well as in historical and future climate change simulations. The main finding of the study seems to be that in the pre-industrial equi-

librium simulations the sensitivity of the modelled NPP to changes in the phytoplankton light attenuation parameter is rather low, while it increases with climate forcing.

While the paper reads well in most parts and is reasonably well structured, both the scientific question that it aims to address and the implications of the main finding could be made more clear. Open documentation of model testing and tuning is clearly desirable and the authors' recommendation to test parameter sensitivities not only in steady-state, but also transient simulations seems reasonable. Yet, I recommend to address several remarks as outlined below before the paper can be published in BG.

The fact that the modelled primary production is sensitive to a change in the value for the light attenuation parameter is not very surprising. What is the scientific basis for the choice of the parameter k_c? Why not a different parameter? What is the relationship to the Chl : C ratio, which is assumed to be a global constant in this study(?) ?

Maybe the authors could address the question what such a sensitivity test actually tells us. Does it tell us something about the real world? Or rather about the model and its applicability? Is this sensitivity specific to UVic, to EMICs? What can we learn from this sensitivity study about the different regimes of phytoplankton growth limitation (light, nutrients)?

It would be really interesting if one could show that for a range of values for a parameter the observed data can be reproduced reasonably well, but that for the same range the sensitivity is very large for a future climate simulation. However, this does not seem to be the case here, since the results seem to illustrate that for no single parameter value the match to observations is convincing. (Lower k_c values seem to fit better to surface chlorophyll, higher ones better to biogeochemical tracer profiles.)

If the main point of the paper is the increasing sensitivity of modelled primary production with increasing forcing to changes in the value of the parameter k_c, then this should be supported more clearly by the shown figures. It is difficult to see how the authors arrive at the conclusion that the sensitivity to k_c is generally modest for the

steady-state. There seems to be quite a spread in some simulated variables.

It would be interesting to see the sensitivity of both the physical and the biogeochemical model part to changes in k_c if the shortwave heating were included.

What do other studies say about the sensitivity of NPP to changes in the light field? What do other studies say about the sensitivity of oxygen to the changes in NPP? This could be further elaborated in the discussion section.

Specific comments:

Introduction: The discussion of underwater light field descriptions seems to be rather long and not taken up in later sections of the paper. Furthermore, Manizza et al (2005) and Gnanadesikan and Anderson (2009) study the effect of including the shortwave heating by phytoplankton and others in their models, and also Kim et al (2015) have the shortwave heating included in their model. This effect is not addressed at all in the rest of the present paper.

Section 2, Methods:

- p3, lines 16-19: As far as I understand, the erroneous calculation of the light attenuation with depth was treated and fixed in Keller et al (2012). It would be helpful to be more specific here and clarify that in the present paper the corrected version of the model as described in Keller et al (2012) is used. In the current manuscript it says that "This calculation is corrected here."

- p3, line 20ff: This sentence is not clear and should be split into at least two parts. The authors should make clear that the sensitivity of biogeochemistry *to* different values of k_c both in a steady-state simulation and in a transient simulation - and not *to* k_c and *to* climate forcing - is assessed. Also, the authors should clarify that the transient simulation is not only a historical one, but also includes a future scenario extending to the year 2300. In addition, the description of the forcing could be more specific (e.g., what are "historical atmospheric CO_2 changes"? Is the model driven by CO_2

concentrations or emissions? What are "agricultural" emissions? $CO_2$ emissions due to land use? Is the model forced by CFC emissions or concentrations? etc.) Should it be changes *in* land ice instead of *to*?

- p3, line 24: replace "models were" with "simulations were" or with "model was"?

- p3, line 25: Is the non-$CO_2$ GHG radiative forcing prescribed or calculated in the model from the concentrations? Or from the emissions? What does "forced using ... fractions of the land surface devoted to agricultural uses" mean? How is the land use forcing realized here?

- p4, line 7: I do not understand this sentence. Replacing the default value of which parameter with which values results in the different shown values for k_c? Do the authors want to say that the 3 given references give different values (or a range) for k_c, or am I missing anything here?

- p.4, line 11: Please clarify what is meant by "any value assigned to k_c is going to be highly model-dependent".

- p4, line 12/13: What conversion factor for Chl : N was used and why?

- The description of the observational datasets could be moved to the methods section, and could be more specific (which tracer data are taken from which dataset).

Section 3, Results:

- p4, line 19ff: I am not sure how the provided Figure 1 showing surface chlorophyll illustrates the sensitivity of primary production. Why don't the authors show simulated (vertically integrated) primary production? Also, the authors could explain why they are comparing satellite data to results from simulations at pre-industrial conditions.

- p4, line 22/23: The authors write that chlorophyll is overestimated in the simulations compared to satellite data in the tropics and the southern hemisphere mid latitudes, but in the plot it is the tropics and the *northern* hemisphere mid latitudes ($\sim$35-70

degrees) that are overestimated. The authors should check the latitude axes of the plotted data.

- p4, line 27ff: Why is the effect of increasing k_c on surface chlorophyll ("biomass" as written in the text is not shown anywhere in the plots) negative in the Southern Ocean, but positive elsewhere? A more detailed explanation of the mechanisms including the vertical distributions and the different regimes of nutrient/light-limitiation in the different regions would be helpful here. Note that the cited study by Kim et al (2015) shows an increase of surface chlorophyll due to the inclusion of light attenuation by colored detrital matter almost everywhere.

- p4, line 32ff: It would be helpful to see the primary production (profiles) to follow the discussion in this paragraph. Also, it should be clarified whether the purpose of this paragraph is to get a better understanding of the primary production sensitivity or of the consequences this sensitivity has on the distributions of the biogeochemical tracers that are shown.

- p5, line 11ff: What do the authors conclude from the fact that K1 fits best for surface chlorophyll (as stated on p.4, line 24), but K4 and higher fit better for the deep ocean biogeochemical tracers when compared to data from SeaWiFS, WOA, and GLODAP?

- p5, line 16: Since the physical response is the same in all simulations, it seems that in the model there is no effect of the underwater light field on temperature, which may be worth mentioning somewhere in the paper.

- p5, lines 24ff: From the given plots it is hard to see a decline in chlorophyll prior to 2100 in the low latitudes. Also, the terms "NPP", "biomass" and chlorophyll seem to be used interchangeably here.

- p5 line 29: Please specify "both of these regions".

- p5, line 30: What about the peak at ∼80S that is decreased from steady-state to 2100?

- p5, line 31: It is hard to see any differences in chlorophyll in K8 between the steady-state shown in Fig1 and the 2100 state in Fig5.

- p6, line 7: Why does chlorophyll decrease from 2100 to 2300 in most simulations north of 40N?

- p6, line 17f: The spreads in the simulations for different times would be easier to compare to each other if the plots used the same scale. Currently Fig1 uses differences to observations and Fig6 uses absolute values. And why is it unsurprising that the spread is increasing in global NPP, but not in the biogeochemical tracers?

- p6, line 20: It is hard to see the increasing spread in the Southern Ocean from the given plots.

- p6, line 23: The sentence "For all biogeochemical quantities, simulated spread at the surface increases with time." seems to contradict the earlier one saying that the spread in the profiles is retained over time.

Section 4, Discussion:

- p7, line 7: I am not sure the results convincingly show that the value of $k_c$ matters little for primary production in the pre-industrial steady-state of the model for values above 0.04 m^2 / mmol N, but matter more for lower $k_c$ values.

- p7, line 13f: Please clarify what is meant by this sentence ("That this is true…").

- p7, line 25ff: Please explain in more detail how this study demonstrates the importance of which mechanism. - This section could benefit from a quick language check. Some words seem to be missing.

Section 5 Conclusions:

- p7, line 29ff: Saying that the sensitivity is substantial also in steady-state seems to contradict to what has been stated above (that it matters little, see p7, line 7).

Do the terms "steady-state", "equilibrium" and "pre-industrial" all refer to the same simulation? The terms could be used more consistently in the paper.

Figure 1: Why are the chlorophyll profiles (probably global means?) shown in the right panel of Figure 1 not discussed in the manuscript? From this plot it seems that the global mean response is an increase of chlorophyll at the surface and a decrease subsurface for increasing values of k_c. Also, in this plot it seems that there are only 3 model layers shown in the upper 200 m. Is this the vertical resolution of the model? If so, it would be worth mentioning such a coarse resolution in a study that is on the vertical distribution of light in the upper water column. Furthermore, the latitude axis of the plotted data should be checked.

Figure 3: Why is the global alkalinity for K8 so different from the other Ks (especially for the deep ocean)?

---

## Referee Comment (RC2) · Anonymous Referee #2 · 18 May 2017

**General Comments**

The manuscript by Kvale and Meissner presents a study exploring the sensitivity of primary production and biogeochemical tracers to the parameter that controls the magnitude of light attenuation by phytoplankton in the Earth System model UVIC. In a steady-state preindustrial simulation the authors demonstrate that primary productivity is relatively insensitive to the choice of parameter value and suggest that low and high latitude productivity respond in different ways to this choice. However the authors then demonstrate that the choice of parameter value leads to significant differences in primary productivity over a transient $CO_2$ forcing experiment. The authors describe a series of feedbacks between oxygen and the nitrogen cycle that occur with weaker

self-shading that be important to consider for past changes in ecosystems and oxygenation.

The findings of the manuscript contributes to a recent body of literature on the issues of calibrating biogeochemical models for the preindustrial ocean and the potential for biogeochemical feedbacks in both past and future climate changes. As such, the findings are significant for our understanding of biogeochemistry and are appropriate for the journal. However, I have one key question about the interpretation of the modelling results that needs resolving before recommending publication.

**Specific Comments**

The authors describe mechanisms for increases in chlorophyll in the Southern Ocean (a weak self-shading effect facilitating greater production) and the increase in the tropics (a strong self-shading effect leading to a decrease in deep photosynthesis and release of nutrients). I think there is an additional factor that has not been discussed which is the change in the distribution of nutrients. The authors describe a general increase in deep ocean concentrations of $PO_4$ and $NO_3$ with weaker light attenuation (Section 3.1 and Figure 3) but do not mention the concurrent decrease in deep Atlantic concentrations. This pattern has been observed previously in biological pump sensitivity studies as a result of increased biological pump efficiency sequestering more nutrients in the deep ocean (Kwon & Primeau 2006; section 5.3 of Kriest et al., 2012; DeVries et al., 2012). This leads to a drop in surface nutrients concentrations in the Atlantic which are transported to the deep Atlantic via deep water formation. High production, particularly in the Southern Ocean, during experiment K1 could therefore shift the balance of nutrients towards the deep ocean from the surface ocean driving differences in production elsewhere purely from these changes in nutrient distribution. Additionally, because of the significance of production in the Southern Ocean in the simulations, there needs to be some discussion of the representation of iron limitation in the model. Because of the relevance of these mechanisms throughout the manuscript, this additional factor needs to be included and preferably quantified in the

manuscript.

The manuscript would benefit from a minor restructuring. The last section of the Methods would be better suited at the end of the Introduction to give a fuller background and to complement the description of the more complex parameterisations. The Discussion also needs to include some caveats/limitations of the study such as whether these results model dependent, whether the nutrient feedback mechanism is a result of using the more simplified parameterisation and what differences one might expect if using the more complex parameterisation.

**Technical Comments**

Page 2, lines 5-25: this discussion of inherent optical properties is interesting but given the focus of the manuscript on the sensitivity of the simpler parameterisation this needs to be integrated better. I suggest at least revisiting these points in the discussion and commenting how the use of inherent properties might alter the results of the manuscript.

Page 2, line 32: I'm not sure what non-algal particles are or where they are derived from, a small description would be useful.

Page 3, line 1: if possible, could you provide some quantitative estimates of production variability when changing other parameters for comparison?

Page 4, line 2: "probably derive" is odd terminology to use here, either state that it is derived from Fasham or remove the mention to Fasham.

Page 4, lines 1-15: some of the text describing the range of parameter values and their assumptions would be better placed towards the end of the introduction after the description of inherent versus apparent optical properties. This would then serve as a good justification for exploring the sensitivity of model to the parameter value following the discussion of inherent optical properties but which are computationally more expensive.

Page 4, line 15: it would help for clarity to explicitly reiterate here that increasing values of Kc represent increasing attenuation of light with phytoplankton biomass and provide a brief description of the experiments including what aspects of sensitivity you are considering, e.g., sensitivity of productivity and biogeochemical tracers.

Page 4, line 31: Kim et al., (2015) find this effect when testing the light attenuation by CDOM rather than phytoplankton biomass. Are these two parameterisations directly comparable? For example, concentrations of CDOM and biomass might respond differently to stratification and therefore affect attenuation differently?

Page 5, lines 3-6: see specific comments, this needs a reference to tracers in the deep Atlantic.

Page 5, lines 10-13: I appreciate the aim is not to find the best parameter value but it would be useful to state the RMSE for the global tracers, and maybe at the basin-scale too, as it would put later results in context (e.g., page 6 line 13, page 7 line 10  18) and allow comparison against other sensitivity studies such as Kriest et al., (2012).

Page 7, lines 25 – 27: It is not clear what "evolutionary trend in light attenuation characteristics by dominant phytoplankton" refers to. I suggest being explicit about which trends in phytoplankton the authors are referring to (i.e., changes in size, appearances of dominant groups such as calcifiers and diatoms). I am not sure that "evolutionary trend" is appropriate here as this is not a specific trait of the individual organisms themselves. The mention of rapid climate change can also be given more context by citing the Paleo-Eocene Thermal Maximum for example (e.g., Norris et al., 2013).

Figure 3: I find interpretation of this figure difficult because the difference from observations is plotted and therefore includes some structural model error as well as differences from the parameter choice. Plotting the actual profiles, as per Figure 6, might be easier to interpret and allow for direct comparison with Figure 6. The legend is also very small and hard to relate to panels in the far bottom right corner. A graded continual colourscale, rather than different discrete colours, would also help for all plots

with K1 to K8 variability (I also find it hard to distinguish some of these colours when they are next to each other on the plot).

Figure 7: please clarify explicitly that the difference plots are K1 – K8 in the figure caption.

**References**

DeVries, T. and Primeau, F. and Deutsch, C., (2012) The sequestration efficiency of the biological pump. Geophysical Research Letters. 39 (13), L13601

Kriest, I. and Oschlies, A. and Khatiwala, S. (2012) Sensitivity analysis of simple global marine biogeochemical models. Global Biogeochemical Cycles. 26 (2), GB2029

Kwon, Eun Young and Primeau, Francois. (2006) Optimization and sensitivity study of a biogeochemistry ocean model using an implicit solver and in situ phosphate data. Global Biogeochemical Cycles. 20 (4), GB4009

Norris, R.D. and Kirtland Turner, S. and Hull, P.M. and Ridgwell, A (2013) Marine ecosystem responses to Cenozoic global change. Science. 3541 (492).

---

## Author Comment (AC1) · 15 Jun 2017

The authors would like to thank the reviewer for their thoughtful comments, which have led to a substantial revision of our manuscript. The Results and Discussion sections in particular have changed significantly. Reviewer comments are shown in black font. Our response is shown in blue font. Changes to the text are shown in red font.

General comments:

The paper "Primary production sensitivity to phytoplankton light attenuation parameter increases with transient forcing" by Kvale and Meissner addresses a topic that is both relevant and within the scope of BG, namely, the sensitivity of simulated marine biogeochemistry to changes in the parametrization of the under water light field under climate change. In particular, the study addresses the sensitivity of marine net primary production (NPP), alkalinity, dissolved inorganic carbon, phosphate, nitrate, and oxygen, as modelled with the University of Victoria Earth System Climate Model (UVIC ESCM) in pre-industrial equilibrium as well as in historical and future climate change simulations. The main finding of the study seems to be that in the pre-industrial equilibrium simulations the sensitivity of the modelled NPP to changes in the phytoplankton light attenuation parameter is rather low, while it increases with climate forcing.

While the paper reads well in most parts and is reasonably well structured, both the scientific question that it aims to address and the implications of the main finding could be made more clear. Open documentation of model testing and tuning is clearly desirable and the authors' recommendation to test parameter sensitivities not only in steady-state, but also transient simulations seems reasonable. Yet, I recommend to address several remarks as outlined below before the paper can be published in BG.

Two sentences have been added to the end of the Introduction (p3, line 14):

The aim of our study is to assess the sensitivity of modelled net primary production to phytoplankton light attenuation parameter value in an ESM using pre-industrial equilibrated, historical, and projected climate forcing. To our knowledge, such a simplistic assessment has not appeared in the peer-reviewed domain despite there being a wide range of phytoplankton light attenuation parameter values currently in use (described in more detail below) and a demonstrated sensitivity of primary production, export, and nutrients in OGCMs and ESMs to how the underwater light field is modelled (described above).

The fact that the modelled primary production is sensitive to a change in the value for the light attenuation parameter is not very surprising. What is the scientific basis for the choice of the parameter k_c? Why not a different parameter? What is the relationship to the Chl : C ratio, which is assumed to be a global constant in this study(?) ?

The scientific basis of the choice of the range in k_c values tested is now moved to the Introduction (p3-4). This same paragraph also describes why k_w is not tested (the light attenuation of water is fairly well constrained). The sentence related to sea ice light attenuation is modified to read (p3 line 21):

The light attenuation of ice parameter is not examined here: any primary production sensitivity to variation in k_i is likely to have effects relegated to the high latitudes.

The addition to the Introduction described for the first point hopefully now clarifies the reason why phytoplankton light attenuation parameter was chosen for the sensitivity test. Two sentences are also added to the end of the Discussion section (p11, line 21):

It is possible that primary production in our model demonstrates similarly increasing sensitivity to other phytoplankton parameters with climate change, and that the sensitivity of NPP to k_c may be damped or magnified by the choice of other parameter values (e.g., the initial value of the photosynthesis-irradiance curve). Exploring the uncertainty associated with multiple parameter manipulations is costly and better left to offline approaches that can objectively and systematically assess the solution space (see review by Schartau et al., 2017), though as far as we know, offline methods for 3-D models are currently restricted to steady-state analysis.

The citation of Siegel et al. (2005) (p3, line 32) is expanded to better explain the relationship between k_c and Chl:

Even the assumption that k_c varies predictably with chlorophyll concentration can be considered highly simplistic because the co-varying constituents might cause this ratio to fluctuate (Siegel et al., 2005).

Also the sentence describing the conversion of k_c from chlorophyll units to nitrogen units is clarified (p4, line 1):

Conversion of these k_c values to carbon and then nitrogen units using Table 4 from Dutkiewicz et al. (2015) and the Redfield C:N ratio used in our model (6.625)...

Maybe the authors could address the question what such a sensitivity test actually tells us. Does it tell us something about the real world? Or rather about the model and its applicability? Is this sensitivity specific to UVic, to EMICs?
The Discussion section has been substantially revised to include discussion of the implications and limitations of our study (p9-12).

What can we learn from this sensitivity study about the different regimes of phytoplankton growth limitation (light, nutrients)?
A new figure (Figure 3) is added that shows how the different regimes shift in three of the simulations. Nutrient and light limitation regimes are now included as a central part of the Results section (3). (from p5, line 10):
South of this region (around 60°S) UVic ESCM primary production transitions

to being light-limited from being nutrient-limited to the north (annually averaged limitation regimes are shown in Fig. 3) and so reducing the self-shading increases primary production in the light-limited regime.

p5, line 24:

Three distinct regional responses to $k_c$ parameter value choice are therefore apparent. In regions that are light-limited, reducing the light attenuation parameter results in higher NPP (Southern Ocean and Arctic). In regions that are nutrient-limited, reducing the light attenuation parameter results in lower NPP when combined with a strong vertical temperature gradient near the surface (tropics and subtropical gyres). In regions that are nutrient-limited and are characterized by a weak vertical temperature gradient near the surface, reducing the light attenuation parameter results in higher NPP (eastern Pacific, western boundary currents).

And is referred to throughout the Discussion section.

It would be really interesting if one could show that for a range of values for a parameter the observed data can be reproduced reasonably well, but that for the same range the sensitivity is very large for a future climate simulation. However, this does not seem to be the case here, since the results seem to illustrate that for no single parameter value the match to observations is convincing. (Lower k_c values seem to fit better to surface chlorophyll, higher ones better to biogeochemical tracer profiles.)

This would be interesting but in practice unlikely because in the case of surface production parameters, any parameter change that results in a change in primary production patterns is going to require re-tuning of the export parameters to regain agreement with nutrient and carbon observations (deep ocean parameters were tuned based on the original production parameters). However I will be addressing this in my next series of experiments in which I apply an optimisation framework to UVic biogeochemical parameters. Also, RMSE have been added to nutrient profile plots for an easier assessment of match to gridded observations.

If the main point of the paper is the increasing sensitivity of modelled primary production with increasing forcing to changes in the value of the parameter k_c, then this should be supported more clearly by the shown figures. It is difficult to see how the authors arrive at the conclusion that the sensitivity to k_c is generally modest for the steady-state. There seems to be quite a spread in some simulated variables.

Figures and associated text have undergone a major revision to better support the main points of the paper.

NPP is now plotted in Figure 1 as depth-integrated and spatially averaged annual means.

Figure 2 now includes a map view of depth integrated NPP for K1-K8 as well as surface phosphate and nitrate K1-K8.

Figure 3 plots nutrient and light limitation regimes for K1, K3, and K8 at years 1800, 2100, and 2300.

Figure 4 has been modified to spatially averaged profiles rather than deviations from observations. This is a more intuitive way of looking at the spread between simulations and was requested by the second Reviewer.

Figure 6 now shows zonally averaged and depth integrated changes in NPP between 2100 and 1800 for each model simulation to demonstrate the different NPP responses to forcing, and depth integrated NPP differences for K1-K8 and K3-K8 at years 1800 and 2300, to demonstrate the increasing model spread with climate forcing.

Figure 7 shows the increasing model spread in nutrients and carbon by profile plots of basin-averaged differences (K1-K8 and K3-K8) at years 1800 and 2300.

It would be interesting to see the sensitivity of both the physical and the biogeochemical model part to changes in k_c if the shortwave heating were included.

Agreed. Unfortunately our model does not consider an impact on heating of phytoplankton biomass. This is now discussed in the Discussion section (p9, line 19):

The Kim et al. (2015) model allowed biological light attenuation to reduce shortwave heating of the water column while our model does not account for this. Including a reduction of near-surface temperatures with strong self-shading might reduce the increase we find in NPP with higher values of $k_c$, though Manizza et al. (2005) found inclusion of a shortwave feedback to NPP can also enhance spring sea surface temperatures and reduce sea ice.

and p11, line 29:

Lastly, the impact of phytoplankton shading on water column heating is not considered here. This is a potentially significant omission with respect to the climate change response of model physics as global net primary production increases strongly in all of our simulations but never contributes to regional cooling, in contrast to the Manizza et al. (2005) finding that light attenuation by biomass can amplify the seasonal cycle of temperature, mixed layer depth and ice cover by about 10% under pre-industrial conditions. From a global perspective, increasing shortwave penetration along the equator can warm regions to the south (Gnanadesikan and Anderson, 2009), which might damp southward nutrient transport in our low-light attenuation simulations by increasing local export production. However, increasing shortwave penetration in the Southern Ocean can enhance mode water formation from subtropical water (Gnanadesikan and Anderson, 2009), which might enhance the runaway nutrient feedback we demonstrate in low-$k_c$ simulations.

What do other studies say about the sensitivity of NPP to changes in the light field?

The discussion of previous studies relating NPP and light parameterisation is expanded (p9 line 13):

This finding somewhat agrees with Kim et al. (2015) who found a decoupling between nutrient concentrations and biomass when light attenuation of CDOM was accounted for in their ESM. Including light attenuation of CDOM (therefore raising total model light attenuation) increased surface nutrient concentrations in their model through a similar mechanism (shoaling of the biomass and production profiles), however they found CDOM light attenuation decreased depth-integrated biomass and attributed the increasing surface nutrients to less total production. Our model demonstrates an increase in depth-integrated NPP with increasing light attenuation. The Kim et al. (2015) model allowed biological light attenuation to reduce shortwave heating of the water column while our model does not account for this. Including a reduction of near-surface temperatures with strong self-shading might reduce the increase we find in NPP with higher values of $k_c$, though Manizza et al. (2005) found inclusion of a shortwave feedback to NPP can also enhance spring sea surface temperatures and reduce sea ice.

and NPP drivers with climate forcing (p10, line 29):

A recent review of the drivers of change in global NPP in a suite of OGCMs and ESMs to which climate forcing was applied found the low latitudes contained the largest spread in model response, with global trends comprising a balance between increasing metabolic rates and increasing stratification (Laufkötter et al., 2015). Our results suggest differences between this balance across models might be partly related to differences in the treatment of phytoplankton light attenuation.

What do other studies say about the sensitivity of oxygen to the changes in NPP? This could be further elaborated in the discussion section.

Oxygen and NPP are now mentioned (p11, line 7):

That oxygen is sensitive to model treatment of NPP (e.g., Kriest et al., 2012), and that Southern Ocean biological processes can affect global nutrient, carbon, and oxygen distributions (e.g., Kwon and Primeau, 2006; DeVries et al., 2012; Kriest et al., 2012; Keller et al., 2016) are not new findings but, as far as we know, our study is the first to demonstrate the potential for denitrification in the Southern Ocean.

Specific comments:

Introduction: The discussion of underwater light field descriptions seems to be rather long and not taken up in later sections of the paper. Furthermore, Manizza et al (2005) and Gnanadesikan and Anderson (2009) study the effect of including the shortwave heating by phytoplankton and others in their models, and also Kim et al (2015) have the shortwave heating included in their model. This effect is not addressed at all in the rest of the present paper.

The omission of shortwave heating sensitivity to light attenuation parameter is now addressed in the Discussion. See page 4 of this document for the relevant edits.

Section 2, Methods:
- p3, lines 16-19: As far as I understand, the erroneous calculation of the light attenuation with depth was treated and fixed in Keller et al (2012). It would be helpful to be more specific here and clarify that in the present paper the corrected version of the model as described in Keller et al (2012) is used. In the current manuscript it says that "This calculation is corrected here."
According to our local documentation of changes to the model code, a newer light bug fix occurred since Keller et al (2012). In our text we refer to an erroneous depth calculation, while the Keller paper refers to a problem with biomass (diazotrophs were not accounted for, and depth-integrated biomass was incorrectly calculated).

- p3, line 20ff: This sentence is not clear and should be split into at least two parts. The authors should make clear that the sensitivity of biogeochemistry *to* different values of k_c both in a steady-state simulation and in a transient simulation - and not *to* k_c and *to* climate forcing - is assessed. Also, the authors should clarify that the transient simulation is not only a historical one, but also includes a future scenario extending to the year 2300. In addition, the description of the forcing could be more specific (e.g., what are "historical atmospheric CO_2 changes"? Is the model driven by CO_2 concentrations or emissions? What are "agricultural" emissions? CO_2 emissions due to land use? Is the model forced by CFC emissions or concentrations? etc.) Should it be changes *in* land ice instead of *to*?
The paragraph has been modified to read (p4, line 21):
Our study examines model biogeochemical sensitivity to a spread of $k_c$ values at both equilibrium in a pre-industrial climate (atmospheric $CO_2$ concentration of 283.8 ppm) and a future projection. We use historical atmospheric $CO_2$ concentrations, agricultural land cover, volcanic radiative forcing, sulphate aerosol and CFC concentrations to force the model, as well as changes in land ice and solar forcing from year 1800 to 2005 following Machida et al. (1995); Battle et al. (1996); Etheridge et al. (1996, 1998); Flückiger et al. (1999, 2004); Ferretti et al. (2005); Meure et al. (2006).

- p3, line 24: replace "models were" with "simulations were" or with "model was"?
Changed to "simulations were"

- p3, line 25: Is the non-CO_2 GHG radiative forcing prescribed or calculated in the model from the concentrations? Or from the emissions? What does "forced

using . . . fractions of the land surface devoted to agricultural uses" mean? How is the land use forcing realized here?

The text is modified to read (p4, line 25):

From year 2005 to 2300 the simulations were forced using increasing $CO_2$ and non-$CO_2$ greenhouse gas concentrations, projected changes to the fraction of the land surface devoted to agricultural uses (calculated to year 2100 by Hurtt et al., 2011, and then held constant after), and changes in the direct effect of sulphate aerosols following "business-as-usual" RCP scenario 8.5 (RCP8.5, Riahi et al., 2007; Meinshausen et al., 2011).

- p4, line 7: I do not understand this sentence. Replacing the default value of which parameter with which values results in the different shown values for k_c? Do the authors want to say that the 3 given references give different values (or a range) for k_c, or am I missing anything here?

The sentence is modified to read (p3, line 29):

Estimates of k_c vary widely: for example, 0.014 m^2(mg Chl a)^{-1} (generally applicable, Lorenzen 1972), 0.041 m^2(mg Chl a)^{-1} (Southern Ocean, Bracher Tilzer 2001), or a range from 0.006 to 0.015 m^2(mg Chl a)^{-1} assuming all phytoplankton represent mixes of specific species of dinoflagellates, calcifiers, or diatoms (Falkowski et al., 1985).

- p.4, line 11: Please clarify what is meant by "any value assigned to k_c is going to be highly model-dependent".

The sentence is modified to read (p3, line 33):

In practice, any value assigned to k_c is going to be highly model-dependent (e.g., 0.058 m^2(mg Chl a)^{-1} in Wang et al., 2008) because of the wide range of observational estimates and the necessary conversion from chlorophyll to model nutrient units, which requires some simplifying assumptions that depend on model structure.

- p4, line 12/13: What conversion factor for Chl : N was used and why? - The description of the observational datasets could be moved to the methods section, and could be more specific (which tracer data are taken from which dataset).

The sentence is modified to read (p 4, line 1):

Conversion of these k_c values to carbon and then nitrogen units using Table 4 from Dutkiewicz et al., 2015 and the Redfield C:N ratio used in our model (6.625) yields a range of 0.008 to 0.054 (m mmol N m^{-3})^{-1} (though higher values in models exist- Evans and Parslow (1985) used a value of 0.12 (m mmol N m^{-3})^{-1}).

Section 3, Results:

- p4, line 19ff: I am not sure how the provided Figure 1 showing surface chlorophyll illustrates the sensitivity of primary production. Why don't the authors show simulated (vertically integrated) primary production? Also, the authors could explain why they are comparing satellite data to results from simulations at pre-industrial conditions.

Chlorophyll plots have been replaced with NPP plots.

- p4, line 22/23: The authors write that chlorophyll is overestimated in the simulations compared to satellite data in the tropics and the southern hemisphere mid latitudes, but in the plot it is the tropics and the *northern* hemisphere mid latitudes (~35-70 degrees) that are overestimated. The authors should check the latitude axes of the plotted data.

This section has been substantially revised to compare NPP between simulations. Reference to chlorophyll is removed.

- p4, line 27ff: Why is the effect of increasing k_c on surface chlorophyll ("biomass" aswritten in the text is not shown anywhere in the plots) negative in the Southern Ocean, but positive elsewhere? A more detailed explanation of the mechanisms including the vertical distributions and the different regimes of nutrient/light-limitiation in the different regions would be helpful here. Note that the cited study by Kim et al (2015) shows an increase of surface chlorophyll due to the inclusion of light attenuation by colored detrital matter almost everywhere.

The impact of light attenuation parameter on different growth regimes is now included in section 3.1 (p5, line 8):

In the Southern Ocean, K1 zonally averaged primary production rates can exceed those of K8 by more than a factor of 3 because phytoplankton in K1 do not self-shade as strongly during the Austral summer, thereby allowing for a stronger seasonal cycle. South of this region (around 60°S) UVic ESCM primary production transitions to being light-limited from being nutrient-limited to the north (annually averaged limitation regimes are shown in Fig. 3) and so reducing the self-shading increases primary production in the light-limited regime. The transition zone between light and nutrient limitation is well-mixed, and lateral advection of regenerated nutrients from the light-limited regime boosts NPP in the nutrient-limited regime in low-kc value simulations. In the more stratified (and nutrient-limited) tropics, the effect is opposite in that K8 yields zonally averaged NPP of up to double K1 because stronger self-shading inhibits deeper photosynthesis (see the globally averaged NPP depth profile plot in Fig. 1, which is dominated by the low latitude response), making more regenerated nutrients available at the surface (Figs. 2 and 4, and similar to the effect of light attenuation by CDOM described previously by Kim et al. 2015). Higher nutrient concentrations at the tropical surface in K8 cause a net increase in depthintegrated primary production because of the temperature dependency of primary production and respiration in the model (the warmer surface increases the production and remineralisation rates, resulting in higher NPP). Simulation differences in the tropical eastern Pacific upwelling region arise from processes similar to those described in the Southern Ocean. While the eastern Pacific upwelling zone is nutrient-limited in our model (like the rest of the tropics, Fig.3), a weak near-surface temperature gradient reduces primary production in the surface layer. Higher light availability in K1 therefore allows for deeper utilization of upwelled nutrients, resulting in higher depth-integrated NPP in K1 compared to K8. Three distinct regional responses to kc parameter value choice are therefore apparent. In regions that are light-limited, reducing the light attenuation parameter results in higher NPP (Southern Ocean and Arctic). In regions that are nutrient-limited, reducing the light attenuation parameter results in lower NPP when combined with a strong vertical temperature gradient near the surface (tropics and subtropical gyres). In regions that are nutrient-limited and are characterized by a weak vertical temperature gradient near the surface, reducing the light attenuation parameter results in higher NPP (eastern Pacific, western boundary currents).

- p4, line 32ff: It would be helpful to see the primary production (profiles) to follow the discussion in this paragraph. Also, it should be clarified whether the purpose of this paragraph is to get a better understanding of the primary production sensitivity or of the consequences this sensitivity has on the distributions of the biogeochemical tracers that are shown.
Primary production profiles have been included in Figure 1. An introductory sentence has been added to the paragraph (p5, line 33):
Carbon and nutrient distributions in the UVic ESCM are also sensitive to $k_c$ because parameter choice affects the efficiency of the biological pump (Fig. 1), leading to a redistribution of nutrients (Fig. 4).

- p5, line 11ff: What do the authors conclude from the fact that K1 fits best for surface chlorophyll (as stated on p.4, line 24), but K4 and higher fit better for the deep ocean biogeochemical tracers when compared to data from SeaWiFS, WOA, and GLODAP?
Reference to chlorophyll is now removed from the manuscript.

- p5, line 16: Since the physical response is the same in all simulations, it seems that in the model there is no effect of the underwater light field on temperature, which may be worth mentioning somewhere in the paper.
This is now mentioned in the Discussion section (see edits on page 4 of this manuscript):

- p5, lines 24ff: From the given plots it is hard to see a decline in chlorophyll prior to 2100 in the low latitudes. Also, the terms "NPP", "biomass" and chlorophyll seem to be used interchangeably here.

Figure 5 has been revised to show NPP changes from years 1800 to 2100 for each simulation, and K1-K8 and K3-K8 differences at years 1800, 2100, and 2300. The decline in NPP in the low latitudes for the lower K simulations is shown. References to chlorophyll and biomass are removed from the discussion, which now focuses on NPP.

- p5 line 29: Please specify "both of these regions".

This sentence was deleted.

- p5, line 30: What about the peak at ~80S that is decreased from steady-state to 2100?

Added to the paragraph (p7, line 22):

Along the Antarctic margin (around 80 S), local freshening causes large local declines in NPP in simulations using weaker self-shading, though the region is not nutrient-limited in our model. The mechanism for the decline is a drop of seawater temperature in the second ocean depth layer, which disproportionately affects simulations that have deeper NPP. Simulations K7 and K8 are relatively less sensitive to increasing stratification (and associated nutrient limitation) because their high $k_c$ values raise primary production higher in the water column, thereby raising surface nutrient concentrations and allowing the phytoplankton to be less reliant on resupply of nutrients from deeper waters.

- p5, line 31: It is hard to see any differences in chlorophyll in K8 between the steadystate shown in Fig1 and the 2100 state in Fig5.

Figure 5 has been revised to show NPP changes from years 1800 to 2100 for each simulation, and K1-K8 and K3-K8 differences at years 1800, 2100, and 2300.

- p6, line 7: Why does chlorophyll decrease from 2100 to 2300 in most simulations north of 40N?

Discussion of chlorophyll is now removed from the text. Northern hemisphere NPP is discussed as (p7, line 28):

All simulations show an increase in NPP north of about 50°N to 60°N, which is driven by increasing light availability and warming temperatures in all light attenuation tests.

That chlorophyll declines while NPP increases is probably due to the enhanced temperatures in this region that raise primary production while not requiring a corresponding increase in biomass.

- p6, line 17f: The spreads in the simulations for different times would be easier to compare to each other if the plots used the same scale. Currently Fig1 uses

differences to observations and Fig6 uses absolute values. And why is it unsurprising that the spread is increasing in global NPP, but not in the biogeochemical tracers?

Figure 6 has been replaced with a new figure showing the K1-K8 and K3-K8 differences at years 1800 and 2300. The sentence referred to has been revised (p8, line 13):

Most biogeochemical quantities retain the pre-industrial spread in global profiles with increasing $CO\_2$ forcing...

- p6, line 20: It is hard to see the increasing spread in the Southern Ocean from the given plots.

Figure 6 has been replaced with a new figure showing the K1-K8 and K3-K8 differences at years 1800 and 2300.

- p6, line 23: The sentence "For all biogeochemical quantities, simulated spread at the surface increases with time." seems to contradict the earlier one saying that the spread in the profiles is retained over time.

Language regarding what we conclude about the level of model sensitivity in steady-state and transient simulations has been revised for clarity throughout the manuscript. The Results section now includes comparison of K3 -K8 in addition of K1-K8 because K1 and K2 have somewhat distinct sensitivity compared to the higher k_c values. This sentence referred to Figure 7, which has been removed from this version of the manuscript.

Section 4, Discussion:
- p7, line 7: I am not sure the results convincingly show that the value of k_c matters little for primary production in the pre-industrial steady-state of the model for values above 0.04 m^2 / mmol N, but matter more for lower k_c values.

The sentence is deleted.

- p7, line 13f: Please clarify what is meant by this sentence ("That this is true. . ."). The sentence is deleted.

- p7, line 25ff: Please explain in more detail how this study demonstrates the importance of which mechanism. - This section could benefit from a quick language check. Some words seem to be missing.

The sentence is revised (p11, line 17):

Our tests demonstrate another potential mechanism for the increase in ocean oxygen inventory in equilibrated conditions as well as for a stabilisation of oxygen under rapid climate change- an evolved increase in light attenuation by dominant phytoplankton, which in our model increases ocean oxygen inventory and mitigates total oxygen change with climate forcing.

Section 5 Conclusions:

- p7, line 29ff: Saying that the sensitivity is substantial also in steady-state seems to contradict to what has been stated above (that it matters little, see p7, line 7). This sentence has been deleted. Language regarding what we conclude about the level of model sensitivity in steady-state and transient simulations has been revised for clarity throughout the manuscript.

Do the terms "steady-state", "equilibrium" and "pre-industrial" all refer to the same simulation? The terms could be used more consistently in the paper. References to "steady-state" have been replaced by "pre-industrial equilibrium" or similar throughout the text.

Figure 1: Why are the chlorophyll profiles (probably global means?) shown in the right panel of Figure 1 not discussed in the manuscript? From this plot it seems that the global mean response is an increase of chlorophyll at the surface and a decrease subsurface for increasing values of k_c. Also, in this plot it seems that there are only 3 model layers shown in the upper 200 m. Is this the vertical resolution of the model? If so, it would be worth mentioning such a coarse resolution in a study that is on the vertical distribution of light in the upper water column. Furthermore, the latitude axis of the plotted data should be checked.
Chlorophyll is now replaced with NPP in the figure. The NPP profiles are discussed in the text (p5, line 14):
In the more stratified (and nutrient-limited) tropics, the effect is opposite in that K8 yields zonally averaged NPP of up to double K1 because stronger self-shading inhibits deeper photosynthesis (see the globally averaged NPP depth profile plot in Fig. 1, which is dominated by the low latitude response),...

The potential effects of model resolution and the depth layers are included in the Discussion section (p9, line 32):
In this particular region, higher vertical resolution might reduce the overall NPP response of the Southern Ocean to decreasing light attenuation parameter by reducing advected regenerated nutrients and reducing preformed nutrients made available for primary production by reduced self-shading. In the stratified low latitudes, higher vertical resolution might reduce the nutrient shoaling effect of strong self-shading.

Figure 3: Why is the global alkalinity for K8 so different from the other Ks (especially for the deep ocean)?
There was an error in the plotting script. This figure has been replaced with full values for a more intuitive comparison to observations.

---

## Author Comment (AC2) · 15 Jun 2017

The authors would like to thank the reviewer for their thoughtful comments and their effort in helping us to improve our manuscript. This latest version represents a substantial revision of the original document, particularly of the Results and Discussion sections. Reviewer comments are shown in black font. Our response is shown in blue font. Changes to the text are shown in red font.

**General Comments**

The manuscript by Kvale and Meissner presents a study exploring the sensitivity of primary production and biogeochemical tracers to the parameter that controls the magnitude of light attenuation by phytoplankton in the Earth System model UVIC. In a steady-state preindustrial simulation the authors demonstrate that primary productivity is relatively insensitive to the choice of parameter value and suggest that low and high latitude productivity respond in different ways to this choice. However the authors then demonstrate that the choice of parameter value leads to significant differences in primary productivity over a transient CO2 forcing experiment. The authors describe a series of feedbacks between oxygen and the nitrogen cycle that occur with weaker self-shading that be important to consider for past changes in ecosystems and oxygenation.

The findings of the manuscript contributes to a recent body of literature on the issues of calibrating biogeochemical models for the preindustrial ocean and the potential for biogeochemical feedbacks in both past and future climate changes. As such, the findings are significant for our understanding of biogeochemistry and are appropriate for the journal. However, I have one key question about the interpretation of the modelling results that needs resolving before recommending publication.

**Specific Comments**

The authors describe mechanisms for increases in chlorophyll in the Southern Ocean (a weak self-shading effect facilitating greater production) and the increase in the tropics (a strong self-shading effect leading to a decrease in deep photosynthesis and release of nutrients). I think there is an additional factor that has not been discussed which is the change in the distribution of nutrients. The authors describe a general increase in deep ocean concentrations of PO4 and NO3 with weaker light attenuation (Section 3.1 and Figure 3) but do not mention the concurrent decrease in deep Atlantic concentrations. This pattern has been observed previously in biological pump sensitivity studies as a result of increased biological pump efficiency sequestering more nutrients in the deep ocean (Kwon & Primeau 2006; section 5.3 of Kriest et al., 2012; DeVries et al., 2012). This leads to a drop in surface nutrients concentrations in the Atlantic which are transported to the deep Atlantic via deep water formation. High production, particularly in the Southern Ocean, during experiment K1 could therefore shift the balance of nutrients towards the deep ocean from the surface ocean driving differences in production elsewhere purely from these changes in nutrient distribution. Additionally, because of the significance of production in the Southern Ocean in the simulations, there needs to be some discussion of the representation of iron limitation in the model. Because of the relevance of these mechanisms throughout the manuscript, this additional factor needs to be included and preferably quantified in the manuscript.

Reference to the named manuscripts has now been included to the paper. The results section now includes mention of this deep ocean /Atlantic relationship (p5 line 34):

Carbon and nutrient distributions in the UVic ESCM are also sensitive to  $k_c$  because parameter choice affects the efficiency of the biological pump (Fig. 1), leading to a redistribution of nutrients (Fig. 4). Low-value  $k_c$  simulations experience a greater proportion of global NPP in the high latitudes (regions with higher sequestration efficiency; DeVries et al. 2012), and increasing the  $k_c$  value shifts NPP towards the tropics (a region of lower sequestration efficiency; DeVries et al. 2012). As a consequence, more nutrients and carbon end up in the abyssal Pacific Ocean in low-value  $k_c$  simulations than in higher value ones. Increased storage of nutrients in this deep ocean basin reduces the inventory available for subduction in the northern Atlantic (e.g., Kwon and Primeau, 2006; Kwon\_et al., 2009; Kriest et al., 2012), where water column concentrations of nitrate and phosphate decline (Fig. 4).

Iron limitation is now treated to a closer examination. How it is implemented is now mentioned in the Methods section (p4, line 15): Iron limitation is accounted for using a seasonally variable mask of dissolved iron concentrations in the upper three ocean layers (Keller et al., 2012).

Growth limitation maps have been introduced as a new figure (Figure 3). Iron is not limiting in this model on an annual-mean basis, though it is represented. This is discussed (p 10, line 3):

Though iron availability is accounted for in the form of a seasonally-variable mask, in our model iron is not a limiting nutrient on an annually-averaged basis. This is in contrast to evidence of iron limitation in the Southern Ocean, North Atlantic, and eastern boundary currents and upwelling systems (see recent review by Tagliabue et al., 2017). More iron limitation of phytoplankton growth in the UVic ESCM might damp the NPP response we show for lower light attenuation simulations in the Southern Ocean and eastern equatorial Pacific. More iron limitation might also mitigate differences in the efficiency of the global biological pump between high and low-value light attenuation parameter simulations results in more efficient export and storage of nutrients in the deep ocean, particularly the abyssal north Pacific (also found by DeVries et al. 2012). Model phosphate is conserved in our simulations, thus larger deep ocean inventories result in lower concentrations in downstream surface and intermediate waters (in qualitative agreement with Kwon and Primeau 2006; Kriest et al. 2012). The effect of enhanced deep nutrient sequestration is most

apparent in Atlantic phosphate and nitrate profiles, where concentrations are lower for lower kc simulations and NPP is not very much higher at the surface, in spite of being a seasonally well-mixed region. If iron was more limiting in the Southern Ocean deep water formation regions, fewer nutrients would be sequestered in the deep Pacific and more would be available to the north Atlantic, raising regional primary production and export (assuming no iron limitation also existed in the north Atlantic). More iron limitation in the low latitudes might furthermore damp the NPP response of higher kc simulations in the thermally stratified tropics, thus increasing nutrient transport poleward and increasing high latitude NPP.

The manuscript would benefit from a minor restructuring. The last section of the Methods would be better suited at the end of the Introduction to give a fuller background and to complement the description of the more complex parameterisations. The Discussion also needs to include some caveats/limitations of the study such as whether these results model dependent, whether the nutrient feedback mechanism is a result of using the more simplified parameterisation and what differences one might expect if using the more complex parameterisation.

The last paragraph of the Methods section has been moved to the end of the Introduction. The Discussion section has been rewritten to include more limitations and greater context with earlier work (p9 to p12).

**Technical Comments**

Page 2, lines 5-25: this discussion of inherent optical properties is interesting but given the focus of the manuscript on the sensitivity of the simpler parameterisation this needs to be integrated better. I suggest at least revisiting these points in the discussion and commenting how the use of inherent properties might alter the results of the manuscript.

The Discussion section has been substantially rewritten to include more limitations and greater context with earlier work.

Page 2, line 32: I'm not sure what non-algal particles are or where they are derived from, a small description would be useful. "Non-algal" has been replaced with "detrital" throughout the paper.

Page 3, line 1: if possible, could you provide some quantitative estimates of production variability when changing other parameters for comparison? The sentence is modified as:

These are modest changes with respect to other production and export parameters (e.g., Kwon et al. 2009 found a 5 Gt C  $y^{-1}$ , or 50%, increase in global carbon export by raising the export transfer efficiency exponent by 0.4), though regional sensitivities are stronger (Kim et al., 2015).

Page 4, line 2: "probably derive" is odd terminology to use here, either state that it is derived from Fasham or remove the mention to Fasham. Reference to Fasham is removed.

Page 4, lines 1-15: some of the text describing the range of parameter values and their assumptions would be better placed towards the end of the introduction after the description of inherent versus apparent optical properties. This would then serve as a good justification for exploring the sensitivity of model to the parameter value following the discussion of inherent optical properties but which are computationally more expensive.

This paragraph has been moved to the Introduction.

Page 4, line 15: it would help for clarity to explicitly reiterate here that increasing values of Kc represent increasing attenuation of light with phytoplankton biomass and provide a brief description of the experiments including what aspects of sensitivity you are considering, e.g., sensitivity of productivity and biogeochemical tracers.

The passage is modified as:

For our test, we employ eight separate simulations using  $k_c = 0.01, 0.02, 0.03, 0.04, 0.05, 0.06, 0.07$ , and 0.08 (m mmol N m-3)-1. Increasing the light attenuation parameter value increases the self-shading effect of the phytoplankton biomass, reducing the amount of light available for photosynthesis. In the following analysis they will be referred to as 'K1-8', as we assess the impact of parameter choice on model net primary production, carbon and nutrient distributions in a model equilibrated to pre-industrial climate conditions and then forced with historical and projected greenhouse gas concentrations.

Page 4, line 31: Kim et al., (2015) find this effect when testing the light attenuation by CDOM rather than phytoplankton biomass. Are these two parameterisations directly comparable? For example, concentrations of CDOM and biomass might respond differently to stratification and therefore affect attenuation differently?

The sentence is modified to read:

In the more stratified (and nutrient-limited) tropics, the effect is opposite in that K8 yields zonally averaged NPP of up to double K1 because stronger selfshading inhibits deeper photosynthesis (see the globally averaged NPP depth profile plot in Fig. 1, which is dominated by the low latitude response), making more nutrients available at the surface (Figs. 2 and 4, and similar to the effect of light attenuation by CDOM described previously by Kim et al. 2015). Page 5, lines 3-6: see specific comments, this needs a reference to tracers in the deep Atlantic.

The paragraph edits are given under "Specific Comments"

Page 5, lines 10-13: I appreciate the aim is not to find the best parameter value but it would be useful to state the RMSE for the global tracers, and maybe at the basin-scale too, as it would put later results in context (e.g., page 6 line 13, page 7 line 10 18) and allow comparison against other sensitivity studies such as Kriest et al., (2012).

RMSE are now included in the figure.

Page 7, lines 25 – 27: It is not clear what "evolutionary trend in light attenuation characteristics by dominant phytoplankton" refers to. I suggest being explicit about which trends in phytoplankton the authors are referring to (i.e., changes in size, appearances of dominant groups such as calcifiers and diatoms). I am not sure that "evolutionary trendKw is appropriate here as this is not a specific trait of the individual organisms themselves. The mention of rapid climate change can also be given more context by citing the Paleo-Eocene Thermal Maximum for example (e.g., Norris et al., 2013).

"Trend" is modified to "increase".

Norris et al (2013) is now cited (p11, line 13):

... the Southern Ocean and the low latitude Pacific in the real world, and 2) light attenuation characteristics of dominant phytoplankton (Katz et al., 2004) and ocean oxygen content (Lenton et al., 2014) and rates of change (e.g., Paleocene Eocene Thermal Maximum Norris et al., 2013) have changed over geologic timescales.

Figure 3: I find interpretation of this figure difficult because the difference from observations is plotted and therefore includes some structural model error as well as differences from the parameter choice. Plotting the actual profiles, as per Figure 6, might be easier to interpret and allow for direct comparison with Figure 6. The legend is also very small and hard to relate to panels in the far bottom right corner. A graded continual colourscale, rather than different discrete colours, would also help for all plots with K1 to K8 variability (I also find it hard to distinguish some of these colours when they are next to each other on the plot).

Figure 3 has been revised to actual profiles, global RMSE with observations, and a larger legend.

Figure 7: please clarify explicitly that the difference plots are K1 – K8 in the figure caption. Figure 7 is removed.

**References**

DeVries, T. and Primeau, F. and Deutsch, C., (2012) The sequestration efficiency of the biological pump. Geophysical Research Letters. 39 (13), L13601 Kriest, I. and Oschlies, A. and Khatiwala, S. (2012) Sensitivity analysis of simple global marine biogeochemical models. Global Biogeochemical Cycles. 26 (2), GB2029

Kwon, Eun Young and Primeau, Francois. (2006) Optimization and sensitivity study of a biogeochemistry ocean model using an implicit solver and in situ phosphate data. Global Biogeochemical Cycles. 20 (4), GB4009

Norris, R.D. and Kirtland Turner, S. and Hull, P.M. and Ridgwell, A (2013) Marine ecosystem responses to Cenozoic global change. Science. 3541 (492).

---

## Author Response (AR2)

The authors thank both reviewers and the editor for making additional time to review a second draft. Review comments have further improved the manuscript and are reproduced here in italic font. Our responses are given below the comment. Changes to the text are now shown in blue in the manuscript.

Report 1

*The revised manuscript has been improved substantially and most concerns have been addressed. Yet, I still recommend to address the points below.*

*p3, l14: The scientific question that the study wants to address could still be articulated more explicitly. The fact that this sensitivity assessment of a collection of output variables to this one input parameter using one EMIC has not been published previously seems insufficient to me.*

A new sentence has been added to the next line to explain the importance of NPP.

*p4, l1: Which values exactly are taken from Table 4 of Dutkiewicz et al. (2015)? Is it the maximum Chl : C ratio? For which phytoplankton group?*
*This conversion of rates, and possibly other parts of the last paragraph of the introduction, seem to fit better into the methods section.*

The sentence is corrected to include more details of what is used from the table. This paragraph is split between the Introduction and Methods sections.

*In the results section: Is the modelled NPP in agreement with observational data (for the historical period / present) and other model data (for pre-industrial, historical, and future), for regional patterns of vertically integrated or surface values, for zonal means, and for profiles? And why is the model data from the pre-industrial (and not the historical) simulation compared to present-day observational data for the biogeochemical tracers?*

The authors are not aware of a gridded annually averaged dataset of observational NPP that would be readily comparable to the model for a historical or modern period. Instead, commonly used gridded products of observational alkalinity, DIC, PO4, NO3 and O2 are compared to the pre-industrial model in Figure 4.  NPP in this version of the UVic ESCM is compared to other models in Kvale et al (2015).

*p6, l23: The sentence about the observations should be moved to the methods section and could be more specific: which tracer data are taken from which dataset?*

The sentence is moved to the Methods section and modified.

*The authors have thoroughly addressed some potential limitations of their study in the discussion section. Yet, I think the authors go a bit too far with their conclusions and could make them more specific.*
*In this context, (p12, l2) please specify in what way the UVic ESCM is "typical" and*

*what "these limitations" are. Is the UVic a typical Earth system model or a typical Earth system model of intermediate complexity (EMIC)? Or does mainly the ocean physical and biogeochemical model part matter?*

The sentence is modified to be more specific.

*p12, l5-9: Strictly speaking, the present study does not really support these claims, but can only support them for the case of (this version of) the UVic ESCM.*

The first sentence of Section 5 has been modified.

*More technical comments:*

*The second and the last sentence in the abstract could be split into two parts each to improve clarity.*

The sentences have been split.

*The first sentence in the discussion could be made more explicit: What is "pre-industrial equilibrium sensitivity"? Maybe rewrite as "The pre-industrial equilibrium simulations demonstrate a sensitivity…".*

The sentence is rephrased as suggested.

*p12, l1: There is no "runaway" feedback here. I suggest to remove the word or replace it by "positive".*

The word is replaced.

*In the figures, are annual means of single years shown or are those multi-year means?*
The annual averages are all single years.

*For Figure 1, please add that in the left plot zonal means(?) are shown and correct the unit (should be m^-2, not m^-3). For the description of the right plot replace "spatially" by "globally" or "horizontally".*

The captions are rephrased and labels fixed.

*For Figure 2 left and Figure 6, please correct the unit (m^-2, not m^-3).*

Done.

*For Figure 4 and 7, please add whether these are annual means / climatologies.*

Done.

Report 2

*Overall I am happy with the revised manuscript, the authors appear to have put a lot of effort into addressing both sets of reviewer comments in detail. The Discussion section is much improved. The manuscript overall is now much more coherent and concise as a whole. I recommend publication with a few very minor suggestions for changes to the text.*

*Page 3, line 20+: This paragraph needs better integration into the Introduction. In particular, the Methods requires some description of the parameter range used in the experiments and the Introduction ideally would finish the aims of the study (I appreciate I suggested moving this from the methods and my apologies for not being clearer the first time around!). My issue with this paragraph is that it contains details that are specific to the use of the model in the study (e.g., see Page 4, lines 1 to 9) and better suited to the Methods but also details that are general to the use of eqn 1 in modeling (e.g., see Page 3, lines 26 to 35) and more relevant to the discussion of eqn 1 in the Introduction.*

The paragraph is now split between the Methods section and Introduction.

*Page 6, lines 26 – 28: The reason for the choice of two different ranges of K values is not particularly clear here but becomes apparent in the following paragraphs. If I understand correctly, K3 – K8 represent the parameter ranges that better represent the pre-industrial tracer observations and K1 – K8 is the full range tested. I suggest rewording this sentence to something along the lines of comparing the differences between the full sensitivity range tested (K1 – K8) and a subset of the range excluding those two runs that cannot reproduce the preindustrial tracers (K3 – K8).*

This sentence has been clarified.

*Page 10, lines 10 – 12: The comment on deep ocean inventories also needs to be included within the Results section when discussing the carbon and nutrient distributions (i.e., pages 5 & 6) as this is a component of the net change in NPP in the pre-industrial steady-state runs alongside the initial changes in NPP driven by the choice of attenuation-parameter. I note this is perhaps less a component for the transient run as the timescales of nutrient re-distribution are generally longer.*

Deep ocean inventory affecting downstream nutrient availability is discussed (now page 6 lines 3-10)

[revised manuscript text omitted]